# The Global Importance of Gas-phase Peroxy Radical Accretion Reactions for Secondary Organic Aerosol Formation

Alfred W. Mayhew<sup>1</sup>, Lauri Franzon<sup>2</sup>, Kelvin H. Bates<sup>3</sup>, Theo Kurtén<sup>2</sup>, Felipe D. Lopez-Hilfiker<sup>4</sup>, Claudia Mohr<sup>5,6</sup>, Andrew R. Rickard<sup>7,8</sup>, Joel A. Thornton<sup>9</sup>, Jessica D. Haskins<sup>1\*</sup>

- 5 Department of Atmospheric Sciences, University of Utah, Salt Lake City, Utah, 84112, United States of America
  - <sup>2</sup> Department of Chemistry, University of Helsinki, P.O. Box 55 (A.I. Virtasen aukio 1), 00014 Helsinki, Finland
  - <sup>3</sup> Department of Mechanical Engineering, University of Colorado, Boulder, Colorado 80309, United States of America
  - <sup>4</sup> Tofwerk AG, Thun, Switzerland.
- <sup>5</sup> Department of Environmental Systems Science, ETH Zurich, Zürich, Switzerland
- 0 6 PSI Center for Energy and Environmental Sciences, 5232 Villigen PSI, Switzerland
  - <sup>7</sup> Wolfson Atmospheric Chemistry Laboratories, Department of Chemistry, University of York, UK
  - <sup>8</sup> National Centre for Atmospheric Science, Department of Chemistry, University of York, UK
  - <sup>9</sup> Department of Atmospheric Sciences, University of Washington, Seattle, Washington 98195, USA.
- 15 \*Correspondence to: Jessica D. Haskins (jessica.haskins@utah.edu)

## Abstract.

Secondary organic aerosol (SOA) is an important class of atmospheric species with influences on air quality and climate. One understudied SOA formation pathway is gas-phase peroxy radical (RO2) accretion reactions, where two peroxy radicals combine to form a dimer species. This work makes use of recent advances in the theoretical understanding of RO2 accretion reactions to assess their contribution to SOA. After evaluation in a chemical box model, a reduced representation of RO2 accretion reactions was added to a global chemical transport model (GEOS -Chem) to assess the contribution to global SOA and the associated radiative impact. The results of this work suggest that RO2 accretion products may comprise 30-50% of particulate matter (PM2.5) in tropical forested environments, and a smaller proportion in more temperate regions like the south-eastern USA ( $\approx$ 5%). This work suggests that biogenic volatile organic compounds (BVOCs) are the main precursors to accretion products globally, but that a notable fraction of aerosol-phase accretion products come from aromatic-derived RO2 and small acyl-peroxy radicals. Contrary to previous assumptions that accretion products are organic peroxides, the box modelling investigations suggest that non-peroxide accretion products (ethers and esters) could comprise the majority of accretion products in both the gas and aerosol phase. This work provides justification for more extensive measurements of RO2 accretion reactions in laboratory experiments and RO2 accretion products in the ambient atmosphere in order to better constrain the representation of this chemistry in atmospheric models, including a greater level of mechanistic chemical representation of SOA formation processes.

# 1 Introduction

Atmospheric aerosol, solid or liquid particles suspended in the atmosphere, has important negative impacts on human health, as well as influencing the Earth's radiative balance. (Forster et al., 2021; Hallquist et al., 2009) Secondary organic aerosol (SOA) has been shown to comprise a substantial fraction of measured aerosol in a range of environments. (Camredon et al., 2010; De Gouw and Jimenez, 2009; Hallquist et al., 2009; Zhang et al., 2007; Ziemann and Atkinson, 2012) Previous work has shown that covalently bonded dimer species, formed either through gas or particle-phase processes, can contribute to SOA mass.(Kenseth et al., 2018; Zhang et al., 2015) One important formation pathway to form dimer species is the gasphase cross-reactions of peroxy radicals (RO<sub>2</sub>). The dimer products of these RO<sub>2</sub> cross reactions are often termed 'accretion 40 products' due to the possibility to form oligomeric products from sequential oxidation. (Hallquist et al., 2009) Evidence of RO<sub>2</sub> accretion products have been found in a range of chamber and flow-tube experiments conducted with a variety of volatile organic compound (VOC) precursors and oxidant conditions. (Berndt et al., 2015; Ehn et al., 2014; Mettke et al., 2023; Peräkylä et al., 2023) Further mechanistic evidence for RO<sub>2</sub> + RO<sub>2</sub> accretion reactions comes from theoretical calculations demonstrating that this can be a competitive fate for RO<sub>2</sub>-RO<sub>2</sub> complexes, as opposed to the formation of free alkoxy radicals (RO) or the formation of alcohols and carbonyls. (Hasan et al., 2020, 2021; Salo et al., 2022; Valiev et al., 2019) Recent experimental and theoretical work has also suggested that in-complex RO fragmentation can result in the formation of ether or ester accretion products with fewer carbon atoms than the sum of the two peroxy radicals, as opposed to the more commonly considered organic peroxides (Figure 1).(Franzon et al., 2024; Peräkylä et al., 2023) The use of the term 'RO<sub>2</sub> accretion products' in this work refers to dimer species with peroxide, ester, and ether linkages formed through RO<sub>2</sub> cross reactions.

RO<sub>2</sub> accretion is a way to potentially rapidly form high mass species, for example two 10-carbon monoterpene RO<sub>2</sub> species may form a 20-carbon dimer as a primary oxidation product. These high mass species are likely to have low volatilities, meaning they will rapidly partition onto or into existing aerosol, or nucleate new particles, to form SOA. (Dada et al., 2023a; Ehn et al., 2014) This is further strengthened by findings that larger, more functionalised RO<sub>2</sub> form accretion products more efficiently. (Berndt et al., 2018) Given the rapid reaction of RO<sub>2</sub> with nitrogen monoxide (NO) in polluted environments with moderate-to-high NO concentrations, (Jenkin et al., 2019) RO<sub>2</sub> cross reactions are expected to be most important in remote regions far from NO emission sources. Furthermore, since RO<sub>2</sub> cross reactions require the bimolecular reaction of two RO<sub>2</sub> molecules, they will be most important in high VOC environments. Remote tropical forested regions, such as Amazonia, central Africa, regions of south-east Asia, and New Guinea, therefore represent likely hotspots for RO<sub>2</sub> accretion reactions due to low NO concentrations and high biogenic VOC concentrations. (Xu et al., 2022)

There have been two previous attempts to include RO<sub>2</sub> accretion reactions, alongside unimolecular RO<sub>2</sub> autooxidation reactions, in global models. Weber *et al.* included a set of RO<sub>2</sub> dimerisation reactions in their development of a mechanism for the formation of Highly Oxidised Molecules (HOMs) in climate modelling applications. (Weber et al., 2020) Secondly,

Xu et al. have presented a representation of HOM formation from autooxidation and RO<sub>2</sub> accretion reactions for monoterpenes and isoprene in the chemical transport model GEOS-Chem.(Xu et al., 2022)

The work presented here aims to build on this previous work and assess the global impact of RO<sub>2</sub> accretion reactions by constructing chemical mechanisms that make use of recent advancements in the theoretical understanding of the formation of RO<sub>2</sub> accretion products, as well as updates to GEOS-Chem's isoprene chemistry. The mechanisms are evaluated in box model simulations constrained with data from the Southern Oxidant and Aerosol Study (SOAS), which took place in the summer of 2013 in a rural area of the south-eastern United States. (Budisulistiorini et al., 2015; Lee et al., 2016) The global importance of such reactions is then further assessed by inclusion in GEOS-Chem, as well as a comparison of GEOS-Chem results to ambient measurements made in Amazonia during the GOAMAZON aircraft campaign in February-March 2014. Our work treats the RO<sub>2</sub> accretion reactions separately from autooxidation reactions, unlike previous work, since the bimolecular vs unimolecular nature of these chemical processes means that they will have different responses to changing VOC concentrations, despite both being important in low-NO environments. By focusing only on bimolecular RO<sub>2</sub> accretion reactions, we can represent the process in more chemical detail and separate the effects of these two different chemical pathways. We also couple the RO<sub>2</sub> accretion products to the aerosol module based on predicted volatility, in contrast to Xu et al., (Xu et al., 2022) which allows a global assessment of the role of RO<sub>2</sub> accretion products in the formation of SOA, as well as investigations into the importance of different formation pathways and VOC precursors for changes to PM<sub>2.5</sub>. The new organic aerosol (OA) formed from this added pathway also allows for the first quantitative assessment of the global radiative impact of this chemical process.

Figure 1. Schematic showing the formation of peroxide, ether, and ester RO<sub>2</sub> accretion products via intermediate complexes of peroxy and alkoxy radicals.

## 2 Methodology

#### 2.1 Mechanisms

v3.3.1 (MCM) and the standard GEOS-Chem chemical mechanism (v14.5.0).(Bey et al., 2001; Jenkin et al., 2015) Throughout this work, the mechanisms without additional RO<sub>2</sub>-RO<sub>2</sub> accretion reactions added are referred to as the "Base MCM" and "Base GEOS-Chem Mechanism". The MCM subset was extracted (mcm.york.ac.uk, last accessed 2025-04-10) for all of the VOCs for which measurements were available during the SOAS campaign (Table S1).

RO<sub>2</sub> accretion mechanisms were produced using a predictive tool recently developed by Franzon *et al.*, designed to interface with the GECKO-A chemical mechanism generator (GECKO-AP).(Franzon et al., 2024) Lists of RO<sub>2</sub> species were extracted from each mechanism and fed into GECKO-AP to predict bimolecular rate coefficients and the accretion product distributions for each radical pair. The rate coefficients were calculated using the empirical RO<sub>2</sub> + RO<sub>2</sub> rate class

The simulations presented in this work make use of several chemical mechanisms based on the Master Chemical Mechanism

parametrization already used in GECKO-A,(Jenkin et al., 2019) which we expect to predict the rate coefficient reasonably well. However, its prediction of the product distribution is highly uncertain, as reliable experimental data of the accretion product yields from RO<sub>2</sub> + RO<sub>2</sub> reactions is very scarce. Previous experimental data has shown that the yield of accretion products increases significantly with the size of the reactant RO<sub>2</sub> species. (Berndt et al., 2018; Frandsen et al., 2025) This can be explained by the increasing strength of the non-covalent interactions in the intermediate alkoxy radical complex (Figure 1) suppressing the dissociation to free alkoxy radicals. (Franzon, 2023; Franzon et al., 2024) There is no good parametrisation for the yield of the H-shift channel, but a negative correlation between computed H-shift rates and intermolecular interaction energies suggests it can be neglected for sufficiently complex RO<sub>2</sub> reactant pairs. (Hasan et al., 2023) Based on these observations, GECKO-AP was designed to estimate this interaction energy for each reactant pair based on the functional groups present, to systematically exclude all weakly-interacting reactant pairs, and to only consider the peroxide and alkoxy decomposition channels for the remaining, strongly interacting reactant pairs. Notably, all pairs containing the common methyl peroxy radical (CH<sub>3</sub>O<sub>2</sub>) are treated as weakly interacting in this parametrisation, and so are excluded from RO<sub>2</sub> accretion reactions. To decrease the computational burden of the simulations, an additional filter was applied to remove RO<sub>2</sub> + RO<sub>2</sub> reactions with rate coefficients below 3.7×10<sup>-13</sup> cm<sup>3</sup> molecule<sup>-1</sup> s<sup>-1</sup>.

Given the lumping of some RO<sub>2</sub> species in the Base MCM and Base GEOS-Chem Mechanism, decisions had to be made as to the most appropriate molecular representation of each RO<sub>2</sub> in GECKO-AP. The assumed structure of the MCM species were represented by the assigned SMILES string in the MCM database, and the SMILES strings assigned to each GEOS-Chem RO<sub>2</sub> are given in Table S2.

Even with the added filters previously described, the large number of RO<sub>2</sub> species in each base mechanism results in a large number of new species being formed from the added accretion reactions. In order to limit the number of new species, and reduce the number of gas-particle partitioning reactions added, the products were lumped according to their molecular mass and saturation vapour pressure estimated using the mean of the Nannoolal and SIMPOL group additivity methods.(Compernolle et al., 2010; Nannoolal et al., 2004, 2008; Pankow and Asher, 2008) An accretion mechanism was produced for the MCM and GEOS-Chem mechanisms with accretion products lumped according to the closest 1 g mol<sup>-1</sup> molecular mass and the closest order of magnitude saturation vapour pressure. These mechanisms are referred to as "MCM-Accr" and "GC-Accr" for the MCM and GEOS-Chem Mechanism respectively. Additionally, a more reduced mechanism was produced for implementation into the GEOS-Chem model by lumping according to the closest 100 g mol<sup>-1</sup> molecular mass. This mechanism is referred to as "Reduced-GC-Accr". Gas-particle partitioning was also treated differently in each of these mechanisms, as is explained in later sections.

Given the anticipated importance of sesquiterpenes in RO<sub>2</sub> accretion chemistry (Dada et al., 2023b), additional sesquiterpene chemistry was added to the GEOS-Chem Mechanism. Furthermore, in order to track the formation of accretion products from the oxidation of certain VOCs, additional RO<sub>2</sub> species and corresponding reactions were added to the mechanism. For example, in the standard GEOS-Chem mechanism, styrene is oxidised by OH to produce a series of oxidation products, without an intermediate RO<sub>2</sub>. AROMRO<sub>2</sub> is formed from this reaction in order to properly account for NO<sub>x</sub> and HO<sub>x</sub>

cycling. However, simply adding RO<sub>2</sub> accretion reactions for AROMRO2 would result in additional organic products being formed from the oxidation of styrene without a concurrent decrease in the existing oxidation products. As such, the oxidation of styrene by OH (along with other, similar reactions) has been separated into two stages (via an intermediate RO<sub>2</sub>, C2BZRO2) to allow for the additional accretion reactions to act as competition for the formation of other organic oxidation products. The non-accretion reactions modified and added to the Base GEOS-Chem Mechanism are listed in Table S3. The full modified mechanism file has been made available as supplementary information (see Code and Data Availability). GECKO-AP, used to determine the RO<sub>2</sub> accretion reactions, often produces small by-product species as a result of in-

complex alkoxy radical fragmentation processes. These by-products were mapped to existing species with the same structures where possible. When no matching species was present in the MCM, the species was assigned to a dummy species with no chemical losses. In the GEOS-Chem mechanisms, unknown side-products were assigned to the existing species, 'LVOC'.

All of the mechanisms used in this work have been made available as supplementary material (see Code and Data Availability) in FACSIMILE format for the box model simulations and KPP format for the GEOS-Chem simulation.

# 2.2 Ambient Measurements

- As discussed further in later sections, there are very few time-resolved ambient observations of multiple RO<sub>2</sub> accretion products from different VOCs against which to compare model results. However, the SOAS campaign provides a useful dataset of a selection of aerosol phase accretion products measured in a high BVOC, low NO environment, where we would expect to observe notable RO<sub>2</sub> accretion product formation. Furthermore, prior work has shown that most of the OA mass during SOAS was very low volatility.(Lopez-Hilfiker et al., 2016)
- Aerosol-phase accretion products were measured during SOAS using an Iodide Chemical Ionisation Mass Spectrometer (I'-CIMS) with a filter inlet for gases and aerosols (FIGAERO), the operational details of which are explained elsewhere. (Lee et al., 2016) The mass of accretion products predicted in the MCM-Accr mechanism were used to identify accretion product signals from the FIGAERO-CIMS data. A comparison is presented between the measured accretion product concentrations and the simulated concentrations in the box models (see "2.3 Box Modelling"). To conduct this analysis, all species for 155 which an accretion product with the same mass existed in the available FIGAERO-CIMS data were selected. Since gasparticle partitioning wasn't considered in the mechanism for non-accretion product species (i.e. species already present in the base MCM), signals were ignored where the average modelled accretion product concentration was less than four times the average non-accretion product concentration for species of the same mass. This filtering process means that our analysis of the FIGAERO-CIMS data here makes use of 41 of the 85 available measured masses, accounting for an average of 33.3% of the measured signal intensity over the campaign. However, we note that FIGAERO-CIMS data were only reported for 160 organonitrate species during SOAS. Additionally, the thermal desorption of aerosol-phase species from the filter inlet can result in some fragmentation of thermally unstable species, reducing the number of accretion products available for

comparison.(Lopez-Hilfiker et al., 2014) OA measurements from this campaign, measured by aerosol mass spectrometry (AMS), were also compared to model results.(Pai et al., 2020)

Given the importance of tropical forested regions in our analysis, we also compare the change in GEOS-Chem predicted OA (see "2.4 Global Modelling") to the AMS measured OA during the GOAMAZON campaign. However, no measurements of RO<sub>2</sub> accretion products were made during this campaign, thus this comparison can only be used to assess the impact of the additional chemistry on model-measurement comparisons of total PM<sub>1</sub> OA.

# 2.3 Box Modelling

190

195

The box modelling results presented in this work were obtained using AtChem2, an open-source box model designed for easy integration with MCM subsets.(Sommariva et al., 2020) The simulations included constraints of NO, NO2, O3, CO, H<sub>2</sub>O<sub>2</sub>, HNO<sub>3</sub>, SO<sub>2</sub>, and 25 additional VOCs from the SOAS campaign which are listed in Table S1. Photolysis rates were calculated using the implementation of the MCM's photolysis parameterisation in AtChem2, and scaled based on the ratio of the measured and calculated NO<sub>2</sub> photolysis rate (J<sub>NO2</sub>) to account for variations in solar irradiation (e.g. from cloud cover). 175 The modelling approach followed previous ambient box modelling work described in Mayhew et al. 2022, in which each day of the campaign is modelled separately to avoid propagated errors. Measurements used as model input are duplicated for each day to create a 48-hour simulation period, with the initial 24-hours being the spin-up period which allows unmeasured intermediate species to reach steady-state concentrations. For analysis, the spin-up period is removed from each simulation and the model data concatenated to produce a time-series that can be compared against measured data. (Mayhew et al., 2022) 180 As in Mayhew et al. 2022, each species in the mechanism was assigned a deposition velocity based on the chemical structure of the compound, and a first order deposition reaction was added for each based on the measured boundary layer height. Finally, a second first-order loss process was added for all species to account for ventilation. The same diurnally repeating loss rate was added for all species in the mechanism and scaled until the model reproduced diurnal average methyl vinyl ketone (MVK) concentrations. MVK was selected as a multi-generation oxidation product for which we are confident in the 185 mechanistic representation of the chemistry and the quality of the measurement, meaning discrepancies between the two can be reasonably assumed to result from the representation of physical loss processes.

Reversible partitioning reactions were added for all RO<sub>2</sub>-accretion products in the MCM-Accr and GC-Accr mechanisms. This reversible partitioning used the protocol outlined in D'Ambro et al. to represent the condensation rate constant ( $k_{cond}$ ) and evaporation rate constant ( $k_{evap}$ ) of RO<sub>2</sub>-accretion products to and from a particle-phase analogue at rates dependent on the measured aerosol surface area, the accretion product mass, and the accretion product volatility (Equation 1-5).(D'Ambro et al., 2017) The diffusion limited gas-to-particle mass transfer coefficient ( $k_{mt}$ ) is given by Equation 3. The aerosol surface area ( $S_a$ ), average aerosol radius ( $r_p$ ), mass concentration of organic aerosol ( $C_{OA}$ ), and ambient temperature ( $T_a$ ) are all constrained to measured values from SOAS in all box model simulations. The saturation concentration for each compound ( $c_i^*$ ) is calculated using Equation 4 based on the estimated saturation vapour pressure ( $v_p$ ), the molecular mass ( $m_r$ ), and the universal gas constant ( $T_a$ ). In the reversible partitioning box models, the gas-phase diffusion constant ( $T_a$ ) is assumed to be

0.1 cm<sup>2</sup> s<sup>-1</sup> and the mass accommodation coefficient ( $\alpha$ ) is assumed to be 0.1, as in D'Ambro *et al.*(D'Ambro et al., 2017) Mean molecular speed ( $\alpha$ ) was calculated according to Equation 5.

$$k_{cond} = k_{mt}S_a \qquad \qquad Equation \ 1$$

$$k_{evap} = k_{mt}S_a \frac{c_i^*}{c_{OA}} \qquad \qquad Equation \ 2$$

$$k_{mt} = \left(\frac{r_p}{D_g} + \frac{4}{\alpha \omega}\right)^{-1} \qquad \qquad Equation \ 3$$

$$c_i^* = \frac{v_p}{RT} m_r \qquad \qquad Equation \ 4$$

$$\omega = \sqrt{\frac{3RT}{m_r}} \qquad \qquad Equation \ 5$$

In order to match the implementation of the formation of aerosol phase compounds in GEOS-Chem, box model simulations using the Reduced-GC-Accr mechanism used a first- order uptake term in place of the reversible partitioning used in the GC-Accr and MCM-Accr mechanisms. The first-order rate constants ( $k_{1stOrd}$ ) are calculated according to Equation 6-7, taken from the first-order uptake of other low volatility compounds already included in GEOS-Chem. In the Reduced-GC-Accr box models,  $D_g$  is calculated according to Equation 7, as opposed to assuming a constant value of 0.1 cm<sup>2</sup> s<sup>-1</sup>, as was done for the reversible partitioning models, in order to match the representation in GEOS-Chem (Equation 7). The air density in molecules cm<sup>-3</sup> is denoted by M. Results from the reversible partitioning models were analysed to determine the saturation vapour pressures at which accretion products were present almost entirely in the gas and particle phases. From this, it was determined that the particle-phase uptake was negligible for species with a saturation vapour pressure above  $1 \times 10^{-4}$  Pa, and that species with a saturation vapour pressure of  $1 \times 10^{-6}$  Pa or lower existed almost entirely in the particle-phase. As such, uptake coefficients ( $\gamma$ ) for each species were assigned according to Table 1.

$$k_{1stord} = \frac{S_a}{\left(\frac{r_p}{D_g}\right) + 2.749064 \times 10^{-4} \times \frac{\sqrt{m_r}}{\gamma * \sqrt{T}}}$$
 Equation 6
$$D_g = \left(\frac{9.45 \times 10^{17}}{M}\right) \times \sqrt{T} \times \sqrt{3.472 \times 10^{-2} + \frac{1}{m_r}}$$
 Equation 7

Table 1. Uptake coefficients,  $\gamma$ , assigned to accretion products depending on the magnitude of the estimated saturation vapour pressure.

| Saturation Vapour Pressure (Pa) | γ     |
|---------------------------------|-------|
| > 1×10 <sup>-4</sup>            | 0.000 |

| 1×10 <sup>-4</sup>   | 0.3333 |
|----------------------|--------|
| 1×10 <sup>-5</sup>   | 0.6667 |
| ≤ 1×10 <sup>-6</sup> | 1.0000 |

# 2.4 Global Modelling

GEOS-Chem v14.5.0 was used to investigate the global relevance of RO<sub>2</sub>-accretion products. (Bey et al., 2001) All simulations were run at 2°x2.5° global resolution with 47 vertical levels. The model was forced using assimilated meteorological fields from the Modern-Era Retrospective analysis for Research and Applications, version 2 (MERRA-2). (Gelaro et al., 2017) The default GFED biomass burning NO<sub>x</sub> emissions were used globally. The optional modified wet deposition scheme from Luo *et al.* was activated for all simulations. (Luo et al., 2019, 2020) Simulations were spun-up for one year prior to the analysis period in order to ensure all species had reached suitable background concentrations. Simulations were performed between 2013-01-01 and 2014-03-23. Annual average analyses use the year of 2013, and detailed analysis of June and December 2013 is also presented in order to assess seasonal changes between hemispheres. The simulation period was also chosen because it encompasses both the SOAS campaign (2013-06-14 to 2013-07-03) and the period of the GOAMAZON wet-season campaign (2014-02-22 to 2014-03-23). The Rapid Radiative Transfer Model for GCMs (RRTMG) module was enabled in all of the GEOS-Chem runs in order to quantify the impact that additional organic aerosol formed from RO<sub>2</sub> accretion reactions had on the aerosol Direct Radiative Effect (DRE).

The presented simulations used the fullchem mechanism using the complex SOA scheme with semivolatile primary OA (POA). As described previously, the implementation of RO<sub>2</sub>-accretion products in GEOS-Chem assumed a 1<sup>st</sup> order uptake for accretion products, with uptake coefficients assigned based on the predicted saturation vapour pressure of each lumped accretion product. Each of the aerosol-phase accretion products is included in the calculation of a new aerosol tracer added to GEOS-Chem, called DISOA. DISOA is also included in the calculation of the OA tracer in GEOS-Chem, meaning the additional DISOA mass is included in calculations of total PM concentrations as well as being passed to the RRTMG module. The addition of DISOA to OA in GEOS-Chem means that this additional OA is included in the model on top of the OA produced by GEOS-Chem's existing parameterisation (Figure S1). This has potential to result in double-counting if the existing parameterisation already accounts for some fraction of RO<sub>2</sub> accretion product SOA (see "3.2 Contribution to PM<sub>2.5</sub> Mass").

In order to prevent the excessive and unrealistic build-up of accretion products, a reaction with OH was added for each of the gas-phase accretion products, with a rate constant typical of oxidised VOC reactions with OH (1×10<sup>-11</sup> cm<sup>3</sup> molecule<sup>-1</sup> s<sup>-1</sup>). The product of these reactions is a duplicate of LVOC, an existing species in GEOS-Chem used to represent low volatility compounds, such as the non-isoprene dihydroxyepoxydiols (IEPOX) products of isoprene hydroxyhydroperoxide (ISOPOOH) oxidation.(Marais et al., 2016) The duplicate species is termed DILVOC, and the corresponding aerosol-phase

uptake product is termed DILVOCOA. Given the uncertainty in the secondary oxidation of these accretion products, DILVOCOA is not included in the calculation of OA by the aerosol module and so is not included in the calculation of total PM, or any presented data, unless specified otherwise. DILVOCOA is also not passed to the RRTMG module as it is not included in the OA metric. It should be noted that the inclusion of DILVOC is separate from the use of LVOC as a placeholder for certain by-products, as explained earlier. In this case, the by-product is assigned to LVOC directly and so will be included in the calculations of OA and PM in the same way that LVOC is in the base model. A sensitivity test was performed to assess the impact of the selected OH reaction rate on the formation of accretion products. Additional simulations were performed with the accretion product + OH reaction rate constant set to 1×10<sup>-12</sup> cm<sup>3</sup> molecule<sup>-1</sup> s<sup>-1</sup> and 1×10<sup>-10</sup> cm<sup>3</sup> molecule<sup>-1</sup> s<sup>-1</sup>. Each of the simulations were spun up for one year, as with the previously described simulations, and the predicted gas- and aerosol-phase accretion product concentrations for the month of January 2013 are shown in Figure S2 and Figure S3, respectively. These sensitivity tests illustrate that the gas-phase accretion products are sensitive to the change in the OH rate constant, varying by over 600ppt between the two orders of magnitude spanned in the sensitivity test (Figure S2). Conversely, the aerosol-phase accretion products only show a relatively small change in concentration of up to around 0.2 µg m<sup>-3</sup> between each simulation (Figure S3). This demonstrates that the choice of OH oxidation rate constant has little impact on the majority of the conclusions regarding SOA formation in this work, since the dominant loss of lowvolatility gas-phase accretion products is to aerosol uptake, rather than photochemical loss.

GEOS-Chem's 'planeflight' diagnostic was used in order to make direct comparisons between the SOAS box modelling and the GEOS-Chem simulation. Planeflight is designed to allow the sampling of a plane flight trajectory through the model at user-specified times, locations, and heights. In order to sample the SOAS ground-site over the course of the campaign, constant location values were provided with an altitude of 1m. This allows the output of species concentrations of all advected species for the appropriate grid box at high time resolution (1 minute). The inability to output aerosol-phase information using the planeflight diagnostic in the current GEOS-Chem version means that the GOAMAZON comparison was made by outputting aerosol properties across all levels in the appropriate region over Amazonia. The flight trajectory was then sampled afterwards during analysis of the model data.

# 3 Results and Discussion

## 3.1 Simulated Accretion Product Concentrations

Figure 2 shows the modelled concentration of accretion products present in the gas and particle -phases present in the SOAS box models as well as high time resolution output for the SOAS grid-box in GEOS-Chem. Each of the box model simulations agree well with one another, demonstrating that the simplified representation of accretion product chemistry in the GC-Accr and Reduced-GC-Accr mechanisms doesn't have a large impact on accretion product concentrations compared to the more detailed benchmark MCM. Both gas and aerosol-phase accretion products show a strong diurnal pattern, with higher concentrations at night, likely a result of greater proportion of RO<sub>2</sub> being lost to reaction with NO. This is highlighted

most clearly by the minimum of gas-phase accretion products occurring in the morning period, when NO concentrations spike at sunrise.

While the GEOS-Chem simulation produces a similar diurnal profile of accretion product concentrations, the values are around double the values from the box models. While this may indicate differences in physical losses between GEOS-Chem and the box models, it can largely be explained by an under-prediction in NO observed in the GEOS-Chem output compared to the box models, which are constrained to measured NO mixing ratios (Figure S4). This will shift the RO<sub>2</sub> fate to allow a greater proportion of RO<sub>2</sub> to undergo accretion reactions as opposed to reacting with NO. Furthermore, lower NO concentrations at night-time results in much higher night-time RO<sub>2</sub> concentrations in GEOS-Chem compared to the box models (Figure S5) as a result of increased VOC oxidation by NO<sub>3</sub>. While NO is underpredicted in GEOS-Chem, total NO<sub>x</sub> is reasonably reproduced, indicating a shifting of the distribution of NO<sub>x</sub> between NO and NO<sub>2</sub>, likely facilitated by an overprediction in O<sub>3</sub> which has also been noted in previous work. (Mayhew et al., 2023; Travis et al., 2016)

Figure S6 and S7 show the predicted OA concentrations over Amazonia during the GOAMAZON campaign in a format matching the results from Pai *et al.* 2020.(Pai et al., 2020) Notably, the overprediction in OA observed in this region in the previous work (using GEOS-Chem v12.1.1) is not observed in our base model simulations (using GEOS-Chem v14.5.0), indicating that subsequent updates to isoprene chemistry and deposition scheme made in the model have improved estimates of OA.(Bates and Jacob, 2019; Luo et al., 2020) Adding additional OA from RO<sub>2</sub> accretion to the base model results in an overprediction of OA, with the median value of the distribution in Figure S6 and the low-altitude concentrations in Figure S7 being around double the measured and base model values. Assuming that the existing empirically-derived parameterisation in the base model is accurately predicting OA concentrations, a doubling of OA when adding accretion reactions implies that close to 100% of the OA observed during the GOAMAZON flight campaign was comprised of RO<sub>2</sub> accretion products. It is likely that the 100% figure is an overestimate given the remaining discrepancies between the base GEOS-Chem OA predictions and the measured concentrations, the uncertainties associated with the predictions of accretion product concentrations, and previous literature exploring other SOA formation pathways in BVOC-rich environments. Nevertheless, this analysis suggests that accretion products comprise a very large fraction of OA measured during the GOAMAZON campaign.

Figure 2. Accretion product mixing ratios in each of the SOAS box model simulations and the GEOS-Chem simulation output. (a) Gas-phase accretion products, (b) Aerosol-phase accretion products. Lines represent the hourly mean mixing ratio and shaded areas show one standard deviation above and below the mean.





Comparison between the SOAS box model results and the measurements made with FIGAERO-CIMS allows for an approximate assessment of the accuracy of the modelled accretion product concentrations. As noted in previous work, the calibration of I-CIMS can vary by orders of magnitude depending on the structure of organic compounds, (Lee et al., 2014; Mayhew et al., 2022) making it difficult to provide an accurate comparison between the modelled and measured accretion products, for which no calibration standards are available. Suspected accretion products detected with the I'-CIMS were calibrated using a maximum sensitivity approach, meaning the reported concentrations are the minimum possible concentration corresponding to the measured signal. This is potentially amplified by the possibility of thermal decomposition of some species, resulting in lower measured concentrations. The ratio of the mean modelled concentration and mean measured concentration was calculated for each identified accretion product signal over the course of the measurement campaign. These campaign-average ratios showed that the mean ratio value was 4.6, the median 1.6, the maximum 37, and the minimum 0.015. 19 of the 41 signals used in this analysis shows an average model-to-measurement ratio over the course of the campaign of less than one, with 14 showing an average ratio of less than 0.5, indicating an underprediction for these species against the maximum sensitivity calibration used. Accounting for the relative abundance of each accretion product, by calculating the total sum of the concentration of selected accretion products in the models and measurements, demonstrates that the modelled concentrations are generally around two times larger than the measurement (Figure 3, Figure S8). The mean ratio over the course of the campaign is much higher than 2, at 14.9, because of a large over-prediction between the 25th and 29th of June. The median ratio is 2.17 and the root-mean-squared error, normalised to the mean measured value (NRMSE), is 1.97. This offset of two times is consistent with the maximum sensitivity calibration and is within the range of sensitivity differences for organic species measured by I<sup>-</sup>-CIMS.(Lee et al., 2014) However, it should be highlighted that the uncertainties in such calibrations preclude an accurate analysis of the model's ability to predict accretion product concentrations.




Figure 3. Diurnal average aerosol-phase concentrations of selected accretion products corresponding to the 41 selected CIMS signals. The red line shows the simulated concentrations from the MCM-Accr model, with the purple line showing measured concentrations multiplied by 2. The lines show the mean for each hour across the campaign, with the shaded regions showing one standard deviation above and below the mean.

Figure 4 shows the simulated accretion product mixing ratios in the gas and aerosol phases in the modified GEOS-Chem simulation. This figure demonstrates that the south-eastern USA (used as the focus for the box modelling) does show globally high accretion product concentrations, but that remote tropical forested regions exhibit the largest concentrations in both the gas and aerosol phases. For example, Amazonia, Central Africa, and Borneo show average gas-phase accretion product mixing ratios of over 0.6 ppb, and aerosol-phase mixing ratios of around 200 ppt. Analysis of accretion product concentrations over Amazonia during the GOAMAZON period (Figure S9) show a similar diurnal profile as the SOAS campaign (Figure 2), with higher concentrations overnight and a minimum in gas-phase concentrations in the early morning, when NO concentrations spike.

Figure 4. Monthly average ground-level gas (a,c), and aerosol-phase (b,d) mixing ratios simulated by GEOS-Chem with added RO<sub>2</sub> accretion reactions in June (a,b) and December (c,d) 2013.

Figure 5a and Figure 5c show the enhancements in PM<sub>2.5</sub> when adding RO<sub>2</sub> accretion products to GEOS-Chem. The largest increases are seen in forested tropical regions, where the additional accretion products contribute over 5 μg m<sup>-3</sup> of PM<sub>2.5</sub>. Smaller enhancements of around 0.5 μg m<sup>-3</sup> are seen in subtropical and temperate regions during their summer season. Figure 5b and Figure 5d show the enhancements displayed in Figure 5a and Figure 5c expressed as a proportion of total simulated PM<sub>2.5</sub>, in order to contextualise the enhanced PM<sub>2.5</sub> against concentrations in the base simulation. However, it should be noted that there may be some potential for double-counting the accretion products by expressing the increase as a percentage in this manner (see "3.2 Contribution to PM<sub>2.5</sub> Mass"). The largest difference when comparing Figure 5a and Figure 5c to Figure 5b and Figure 5d is the small percentage increase in PM<sub>2.5</sub> seen over central Africa compared to the large


absolute increase. This is the result of high PM<sub>2.5</sub> concentrations in this region in the base model due to biomass burning and high dust concentrations.



Figure S10 shows the change in PM<sub>2.5</sub> averaged over the first 10 vertical levels in the model, equating to around 1.3 km altitude, in order to approximate boundary layer concentrations. Given the strong vertical profile of the RO<sub>2</sub> accretion products, the concentrations are roughly halved by accounting for non-surface grid-boxes. Comparing to results from Xu et al. shows that our simulations generally produce lower aerosol-phase accretion product concentrations than the accretion product concentrations displayed in their simulations.(Xu et al., 2022) Xu et al. do note that their predicted concentrations are likely an upper estimate as they do not account for gas-phase and particle-phase accretion products separately. Finally, it is possible that an inclusion of autooxidation in this analysis, as was done in Xu et al., could increase the mass of accretion product SOA formed by producing heavier, more oxidised RO<sub>2</sub> species that are more likely to participate in accretion reactions and also form lower-volatility products that will be more likely to partition into the condensed-phase. However, we note that our results show including autooxidation would only serve to further exacerbate GEOS-Chem's overprediction of SOA during the GO-AMAZON campaign. Ultimately, this comparison demonstrates that our findings are in-line with previous work despite the larger number of RO<sub>2</sub> species included in our analysis.

Figure 5. The change in monthly average ground-level PM<sub>2.5</sub> concentrations when adding RO<sub>2</sub> accretion reactions to GEOS-Chem. (a) and (c) show the absolute change in ground-level PM<sub>2.5</sub> mass concentration in June and December, respectively. (b) and (d) show the change as a percentage of ground-level PM<sub>2.5</sub> in the base model in June and December, respectively.

## 3.2 Contribution to PM<sub>2.5</sub> Mass

Figure 6 shows the proportion of measured OA estimated to be comprised of RO<sub>2</sub> accretion products in the MCM-Accr SOAS box model, by dividing the modelled accretion product concentrations by the measured OA mass concentration. The highest average contribution of around 6% is observed during the night-time when absolute concentrations are highest, and this proportion regularly peaks to over 10%.

Figure 6. Proportion of measured organic aerosol estimated to be comprised of accretion products in the MCM-Accr SOAS box model simulations, calculated by taking the ratio of the modelled accretion product concentration and the AMS measured OA. Lines represent the hourly mean mixing ratio and shaded areas show one standard deviation above and below the mean.

As previously mentioned, adding RO<sub>2</sub> accretion products to the PM<sub>2.5</sub> metric produced by GEOS-Chem may result in some amount of double-counting of PM<sub>2.5</sub> mass. This is because much of the SOA produced by GEOS-Chem relies on empirically derived SOA yields from precursor VOCs. If RO<sub>2</sub> accretion reactions were properly accounted for in these SOA yields then they would already contribute to the PM<sub>2.5</sub> mass in GEOS-Chem, even if the mechanistic basis for the formation of this portion of SOA wasn't known. Recent work has suggested that many chamber experiments used to inform empirical SOA yields may not accurately-represent real-world atmospheric conditions, including the conditions suitable for RO<sub>2</sub> accretion reactions, and so the extent of double counting in the presented GEOS-Chem simulations is unknown.(Kenagy et al., 2024) In order to assess the sensitivity to potential double-counting, the proportion of PM<sub>2.5</sub> comprised of accretion products was calculated. Notably, this is different from the data presented in Figure 5b and Figure 5d, because no reference to the base model is made in these calculations. Figure 7a shows the values calculated by taking the mass concentrations of aerosol-phase accretion products divided by the total PM<sub>2.5</sub> mass (Equation 8), which is representative of no double counting. Figure 7b assumes that 100% of the accretion products are double counted and so is calculated according to Equation 9, where the mass of aerosol-phase accretion products is subtracted once from the PM<sub>2.5</sub> mass as it is assumed that these accretion products are already accounted for by GEOS-Chem's existing SOA formation parameterisation. Assuming total double

counting will result in a higher proportion of PM<sub>2.5</sub> being comprised of accretion products since the denominator of the fraction in Equation 9 will decrease when the double-counted PM<sub>2.5</sub> is subtracted.

$$\frac{Accr_{aer}}{PM_{2.5}} \times 100 \%$$
 Equation 8 
$$\frac{Accr_{aer}}{PM_{2.5} - Accr_{aer}} \times 100 \%$$
 Equation 9

10




Figure 7. Proportion of PM<sub>2.5</sub> comprised of RO<sub>2</sub> accretion products, calculated using annual average surface concentrations. (a) Simulated accretion product concentrations divided by the simulated PM<sub>2.5</sub> mass concentration (Equation 8). (b) Assumes 100% double counting of accretion products by subtracting the accretion product concentrations from the PM2.5 denominator (Equation 9).

When assuming no double counting, around 30% of annual averaged surface PM<sub>2.5</sub> in tropical regions such as the Amazon, Borneo, and New Guinea is comprised of accretion products (Figure 7a). This increases upon assuming all accretion products are double-counted, to around 50% (Figure 7b). Other regions generally show smaller proportions, and the potential impact of double-counting is also smaller in these areas as a result. The annual average proportion in the south-eastern USA is around 3%, which is lower than the box modelling results presented earlier for the SOAS campaign, but the strong seasonal variation in accretion product contribution to PM<sub>2.5</sub> observed in this region means that summer-time values are closer to 5%, matching the box model results (Figure S11). As such, given the underlying assumptions used in the model representation, we estimate that accretion products contribute between 30-50% of the total annual average PM<sub>2.5</sub> mass in tropical forested regions where this chemistry is promoted.

Figure S12 shows the same proportional contribution figure for OA instead of total PM<sub>2.5</sub>. In this case, some tropical regions show proportional contributions of accretion products to OA of around 50% to 100% assuming 0% and 100% double counting, respectively. The previous analysis of GOAMAZON data suggests that a large proportion of observed OA could come from accretion products (see "3.1 Simulated Accretion Product Concentrations"), this therefore implies that the level of double counting is very high, at least in these regions. As such, the higher proportional contribution estimate of around

50% of total PM<sub>2.5</sub> is more likely to be correct. The large degree of uncertainty in this comparison further highlights the need for more field and laboratory studies to constrain the potential formation of accretion products in high VOC, low NO environments.

As noted in the Methods section, additional OH losses were added for all gas-phase accretion products to produce a chemical species termed DILVOC. Although this species did have an aerosol uptake process included in the mechanism, it was not included in the calculation of PM<sub>2.5</sub> in the adjusted GEOS-Chem simulations due to large uncertainties in understanding around the losses of RO<sub>2</sub> accretion products. However, given the generally low volatility of the accretion products, it is reasonable to assume that their oxidation products would also contribute somewhat to SOA, provided the oxidation didn't result in extensive fragmentation. Including DILVOC in the calculation of the proportional contribution of accretion products to PM<sub>2.5</sub> shows a higher contribution of 45% to 80% over Amazonia (Figure S13). Given the potential significance of these unconstrained reactions for SOA formation and composition, further research is clearly needed to understand the later generation chemical losses of accretion products.

# 3.3 RO<sub>2</sub> Fate







RO<sub>2</sub> cross reactions (both those that form accretion products and non-accretion reactions already in the base GEOS-Chem mechanism that form monomeric products) will be most prevalent in atmospheres where competition for RO<sub>2</sub> loss, predominantly by reaction with NO or HO2, is low. Unimolecular RO2 reactions are included in the GEOS-Chem mechanism for isoprene RO<sub>2</sub>, and will also compete against RO<sub>2</sub> cross reactions under conditions where the RO<sub>2</sub> species have a long bimolecular lifetime. Figure 8 and Figure S14 show the proportional loss of a first-generation isoprene-derived RO<sub>2</sub>, IHOO1 (C<sub>5</sub>H<sub>9</sub>O<sub>3</sub>), to various reaction pathways throughout the month of June and December in the GEOS-Chem simulations as a case study. Given that the data in panels (b-f) are expressed as a proportional loss, the values are independent of RO<sub>2</sub> concentration. Figure 8a shows the total loss rate taken from the model, which is dependent on the concentration of the RO<sub>2</sub> and any reaction partner (e.g. NO, HO<sub>2</sub>, and RO<sub>2</sub>), illustrating regions in which the RO<sub>2</sub> chemistry is most globally important. As expected, reaction with NO generally dominates over land, with the exception of extremely remote regions like rainforest and desert regions, where the average proportion of IHOO1 reacting with NO drops to below 20%. The Sahara sees unimolecular RO<sub>2</sub> losses comprising around half of the IHOO1 fate in June, similar to remote oceanic environments, though IHOO1 concentrations are extremely low in these environments. Reaction with HO2 comprises the largest proportion of RO<sub>2</sub> loss at extreme latitudes during the corresponding summer month, as a result of high levels of solar irradiation, but can also contribute substantially to RO<sub>2</sub> loss in other regions. The proportion of IHOO1 undergoing RO<sub>2</sub> cross-reactions to form accretion products is in agreement with the distribution of accretion product concentrations. Figure 8 and Figure S14 shows a high proportion of IHOO1 undergoing accretion reactions over remote tropical forested regions, such as the Amazon rainforest, where BVOC emissions are high. However, both non-accretion and accretion RO2 cross reactions are also high in remote ocean environments, particularly in winter at extreme latitudes, as a result of lower losses to reaction with HO<sub>2</sub>. Previous measurement and modelling studies have indicated that marine emissions of isoprene and other BVOCs, a source not currently represented in GEOS-Chem, could be substantial and an important precursor to marine SOA.(Ferracci et al., 2024; Rodríguez-Ros et al., 2020; Zhang et al., 2025; Zhang and Gu, 2022) Given the relatively high fraction of RO<sub>2</sub> lost to accretion reactions in these remote marine environments, increasing marine isoprene emissions could result in the efficient formation of SOA from RO<sub>2</sub> accretion products.




Figure 8. (a) Average total loss rate of the primary isoprene RO<sub>2</sub>, IHOO1, during June of the GEOS-Chem simulation. (b-f) Average fractional loss of IHOO1 to each loss pathway, independent of RO<sub>2</sub> concentration.

Figure 9 shows the average fractional loss of a series of RO<sub>2</sub> categories to accretion reactions in June (December data is shown in Figure S15). The categories are: all RO<sub>2</sub>; a primary aromatic RO<sub>2</sub> from styrene and ethyl benzene (C2BZRO2, see "2.1 Mechanisms"); a primary α-pinene RO<sub>2</sub> (APINO2); and a primary sesquiterpene RO<sub>2</sub> (SQTO2, see "2.1 Mechanisms"). The panels in Figure 9 are analogous to Figure 8f for IHOO1, though it should be noted that unimolecular losses are not included for any of the listed RO<sub>2</sub> which will result in an overestimation of the proportion of RO<sub>2</sub> lost to accretion reactions if unimolecular reactions are occurring in the real world. The accretion reactions comprise a small proportional of the total RO<sub>2</sub> loss (Figure 9a) since the C<sub>1</sub> methylperoxy radical (MO2), which is the most prevalent RO<sub>2</sub>, doesn't participate in any of the added accretion reactions due to slow predicted rates.(Franzon et al., 2024) This demonstrates the importance of RO<sub>2</sub> accretion reactions in SOA formation despite the potential for RO<sub>2</sub> accretion reactions to comprise a small proportion of total RO<sub>2</sub> loss. In contrast, a high proportion of C2BZRO2, APINO2, and SQTO2 are lost to accretion reactions (Figure 9b-d).

The higher mass of these C<sub>8</sub>-C<sub>15</sub> RO<sub>2</sub> species compared to IHOO1 better facilitates accretion product formation, but also means that the lower volatility-accretion products are more likely to partition into the aerosol-phase and contribute to SOA.

Figure 9. Proportional loss of RO<sub>2</sub> to accretion reactions, independent of RO<sub>2</sub> concentration, during June of the GEOS-Chem simulation. (a) average for all RO<sub>2</sub>, (b) primary RO<sub>2</sub> from styrene oxidation (C2BZRO2), (c) primary  $\alpha$ -pinene RO<sub>2</sub> (APINO2), (d) primary sesquiterpene RO<sub>2</sub> (SQTO2).

#### 3.4 Accretion Product Characteristics and SOA Composition


Previous investigations have generally assumed that RO<sub>2</sub> accretion products take the form of organic peroxide species.(Berndt et al., 2018; Murphy et al., 2023; Tomaz et al., 2021; Xu et al., 2022; Zhao et al., 2018) However, there is

evidence that the monomer units can instead be linked by ester, ether, and even alkyl functional groups. (Peräkylä et al., 2023) GECKO-AP, used in this work to predict the formation of accretion products, accounts for the potential fragmentation or intramolecular H-shift of one in-complex alkoxy radical, thus allowing for the potential formation of ester and ether products as well as peroxides. (Franzon et al., 2024; Peräkylä et al., 2023) It should be noted that real-world alkoxy radical fragmentation could occur for both of the in-complex radicals, resulting in an alkyl-linked dimer, but this is currently omitted from the GECKO-AP code because the mechanism for their formation is poorly understood. Figure 10a shows the functional groups present in the RO<sub>2</sub> accretion products in the gas and particle phases in the MCM-Accr SOAS box model simulation. The structure of accretion products will depend on the structure of the contributing peroxy radicals, which in turn depends on the VOC mixture present in a given environment. The average diurnal mixing ratios of constrained VOCs during in the MCM box model simulations are shown in Figure S16. Despite the common assumption that RO<sub>2</sub> accretion products are organic peroxide species, over 80% of the gas-phase accretion products during the SOAS campaign are anticipated to be ester or ether species. A larger proportion of aerosol-phase accretion products are peroxides (Figure 10b), compared to the gas-phase, but esters and ethers still comprise the majority. This is consistent with ester and ether products being formed via fragmentation of one RO<sub>2</sub>, resulting in lower mass, more volatile, accretion products.

Figure 10. Hourly mean proportion of accretion products with ester, ether, or peroxide functionality in the MCM-Accr SOAS box model. 'Multiple' includes accretion products with any combination of the three functional groups. 'None' refers to products with none of the functional groups (due to unstable product radicals, as discussed in the Methods section). (a) shows the composition of gas-phase accretion products, (b) shows the composition of particle-phase accretion products.

The large proportion of ether and ester products is important due to the expected longer atmospheric lifetime of these non-peroxide species, allowing them to be transported further from their source and changing the atmospheric impact of RO<sub>2</sub> accretion reactions. The O-O bond in organic peroxides is generally considered to be relatively weak resulting in atmospheric lifetimes of minutes to days. (Wang et al., 2023) This differential behaviour in atmospheric lifetimes depending

on functionalities is not represented in the chemical mechanisms used in this work, which would likely further reduce the contribution of peroxide species shown in Figure 10.






Figure 11 shows the RO<sub>2</sub> species most important for accretion product formation in the gas and aerosol phases, grouped by their precursor VOC, in the MCM-Accr SOAS box model simulation. The explicitly listed precursor groups in this analysis are "Isoprene", "Monoterpenes", and "Aromatics". Any RO<sub>2</sub> from a single VOC that do not fit these categories are classed as "Other VOC". "Mixed VOC" refers to RO<sub>2</sub> that can be formed in the mechanism from more than one of these groups. Finally, due to their diverse sources, any RO<sub>2</sub> with fewer than 4 carbon atoms, regardless of possible VOC source, is placed in a separate category termed "Small VOC". Figure 11 shows that BVOCs (isoprene and monoterpenes) dominate both the gas-phase and aerosol-phase accretion product precursors, as anticipated in this region. Furthermore, although gas-phase accretion product precursors are reasonably varied, with a large contribution from isoprene and mixed VOCs, the majority of aerosol-phase accretion products come from pure monoterpene RO<sub>2</sub> reactions, with the remainder being largely comprised of monoterpene RO<sub>2</sub> reacting with other categories. This highlights that despite the lower monoterpene concentrations in this location compared to isoprene, the lower volatility of monoterpene-derived accretion products makes them more important when considering SOA formation by this pathway. It should be noted that, due to a lack of measurements, sesquiterpenes were not included in the SOAS box models, hence they are not represented in Figure 11.

Figure 11. Proportional contribution of different VOC precursor combinations to gas-phase (a) and particle-phase (b) RO<sub>2</sub> accretion product concentrations in the MCM-Accr SOAS box models. RO<sub>2</sub> are assigned to categories based on their original precursor VOC. Available categories are "Isoprene", "Monoterpenes", "Aromatics", and "Other VOC". "Mixed VOC" refers to RO<sub>2</sub> that can form from more than one of these groups. Due to their diverse sources, any RO<sub>2</sub> with fewer than four carbon atoms are split into a separate category called "Small VOC".

Figure 12 shows the average VOC precursors that produce gas-phase and particle-phase accretion products in the GEOS-Chem simulation across different regions. As is seen in Figure 11, BVOCs comprise the majority of VOC precursors for both gas-phase and particle-phase accretion products in most environments. Isoprene plays a larger role in gas-phase accretion product formation, whereas monoterpenes and sesquiterpenes show a larger contribution to aerosol-phase accretion product formation. Figure 12 indicates that Small VOCs (RO<sub>2</sub> with 3 or fewer carbon atoms) are the dominant source of gas-

phase accretion products and can also contribute to SOA formation when combining with large RO<sub>2</sub>, such as those from sesquiterpenes. The main contributor to the "Small VOCs" category is the peroxyacetyl radical (CH<sub>3</sub>C(=O)O<sub>2</sub>), called MCO3 in the GEOS-Chem mechanism. This is consistent with previous suggestions that acyl peroxy radicals could be important for the formation of accretion products in SOA.(Franzon et al., 2024; Zhao et al., 2022)

Interestingly, aromatics contribute to 5-10% of globally averaged aerosol-phase accretion products in the GEOS-Chem simulation. Biomass burning releases large amounts of gas-phase oxygenated aromatics to the atmosphere, but anthropogenic emissions can dominate in urban environments. Figure 12 shows that aromatics can comprise a larger proportion of accretion product SOA precursors in heavily-polluted urban environments like Beijing. The 'Other' category in the Beijing aerosol-phase accretion product bars is mostly comprised of "Small VOCs + Aromatics", with the main contributing RO<sub>2</sub> species being MCO3 and the primary xylene RO<sub>2</sub>. This further highlights the importance of small acyl peroxy radicals in accretion product formation, as well as aromatic VOCs, particularly when considering local SOA formation in polluted environments.(Wang et al., 2020)

Figure 12. Proportional contribution of different VOC precursor combinations to gas-phase (a) and particle-phase (b) RO<sub>2</sub> accretion product concentrations in the June and December GEOS-Chem simulations. Data is calculated using the mean of all ground-level gridboxes within the region. RO<sub>2</sub> are assigned to categories based on their original precursor VOC. Available categories are "Isoprene", "Monoterpenes", "Sesquiterpenes", "Aromatics", "Medium VOCs" (RO<sub>2</sub> with more than 3 carbon atoms not belonging to another category), "Small VOCs" (RO<sub>2</sub> with 3 or fewer carbon atoms not belonging to another category). Regions correspond to the following latitude-longitude bounds: Global = (-90.0, -180.0) to (90.0, 180.0); Amazon = (-11.0, -73.0) to (1.0 -57.0); Beijing = (38.0, 115.0) to (40.5, 119.0); West Europe = (45.0, 0.0) to (49.0, 9.0); South-east USA = (32.0, -88.5) to (33.5, -87.0); Alberta = (49.2, -119.2) to (59.8, -110.5).

# 3.5 Impact on Direct Radiative Effect



One of the motivations for studying the formation of SOA is the potential impact on Earth's radiative balance. GEOS-Chem's RRTMG module was used to assess the magnitude of the impact that RO<sub>2</sub> accretion product SOA could have on the Earth's radiative budget. To do this, the direct radiative effect (DRE) was calculated by summing the flux of longwave and shortwave radiation attributed to OA at the top of the atmosphere. (Heald et al., 2014) Given the sign convention of the radiation fluxes, this is equivalent to calculating the difference between incoming short-wave radiation and outgoing longwave radiation at the top of the atmosphere. Then, the difference in DRE was calculated between the base GEOS-Chem model and the model with RO<sub>2</sub> accretion reactions added in order to assess the amount of OA DRE attributable to RO<sub>2</sub> accretion products.

When calculated as a global average over the course of a year, accounting for the surface area of each GEOS-Chem grid box, the change in DRE is only -0.033 W m<sup>-2</sup>. However, the localised behaviour of the simulated PM<sub>2.5</sub> results in a larger decrease in DRE over regions where the PM<sub>2.5</sub> increase is large, reaching an annual average decrease of 1.03 W m<sup>-2</sup> over Amazonia (latitude-longitude bounding box of (-11.0, -73.0) to (1.0 -57.0)), with a standard deviation of 0.15 W m<sup>-2</sup> (Figure 13). This demonstrates that while the RO<sub>2</sub> accretion pathway may not be substantial when considering global radiative balance, it could be an important consideration for investigations into local radiative effects in tropical forested regions. As discussed in previous sections, there is potential for the oxidation products of gas-phase accretion products to contribute to SOA formation, however these products are omitted from this analysis as DILVOCOA is not passed to the OA mass (see "2.4 Global Modelling")"

Figure 13. Calculated difference in annual average Direct Radiative Effect (DRE) when RO<sub>2</sub> accretion reactions are included and excluded from the GEOS-Chem simulations.

# 570 4 Conclusions and Future Work



Comparison of the box model results to measurements indicate that the RO<sub>2</sub> accretion product mechanisms developed here are able to reproduce observed particle-phase concentrations within the limitations of available data. Moving forward, it is clear that extensive measurements of accretion products are urgently needed in both field campaigns and laboratory experiments to better constrain model representations of RO<sub>2</sub> accretion product formation.

Given the lack of measurements, there are a number of key uncertainties that should be addressed to allow for more accurate future investigations. Firstly, the ability to measure these highly oxidised organic species is hampered by the lack of authentic standards and the variable calibration factors for species measured using I<sup>-</sup>-CIMS. Furthermore, the lack of isomerresolved detection possible with mass spectrometric techniques presents issues when analysing complex mixtures of accretion products. The sensitivity tests described in "2.4 Global Modelling" demonstrate that our results regarding the formation of SOA are reasonably insensitive to gas-phase photochemical losses of accretion products, meaning that accurate volatility predictions and a good understanding of the gas-particle partitioning of accretion products is critical to improving the representation of this SOA formation pathway in future investigations. Finally, GECKO-AP includes multiple

simplifications due to a lack of experimental understanding and to limit computational complexity that could be improved in future work, namely the decision to only consider peroxide formation and alkoxy decomposition channels, and to only consider alkoxy decomposition for one of the alkoxy radicals involved.






The complexity of our representation of accretion reactions precluded the inclusion of autooxidation processes, which could be expected to either increase or decrease the anticipated importance of RO<sub>2</sub> accretion reactions. While autooxidation processes will compete with bimolecular RO<sub>2</sub> accretion reactions, they also have the potential to form high molecular weight peroxy radicals that could be more likely to undergo subsequent accretion reactions and form low volatility compounds that contribute to SOA. On the other hand, GECKO-AP also includes the potential for hydroperoxide functional groups, which are readily formed by autooxidation, to inhibit accretion reactions through fragmentation to form OH and a closed-shell organic carbonyl species.(Franzon et al., 2024) Furthermore, there are additional RO<sub>2</sub> loss processes not currently represented in the MCM or GEOS-Chem that may compete with RO<sub>2</sub> accretion reactions and reduce their importance relative to the predictions made in this work. For example, recent work has demonstrated the ability for peroxy radicals to undergo surface redox reactions. (Durif et al., 2024)

Despite the large uncertainties in the RO<sub>2</sub> accretion reactions, our model results suggest that RO<sub>2</sub> accretion products are likely to comprise a large proportion of total PM<sub>2.5</sub> in tropical forested regions, with the potential of comprising the majority of PM<sub>2.5</sub> constituents. Our results suggest that the contribution of accretion products to OA is lower in more temperate and polluted regions, but may still comprise an average of around 5% in the summer time. Consequently, analysis of DRE suggest that this SOA formation pathway could have implications for local radiative balance in tropical regions. Taken together, this implies RO<sub>2</sub> accretion processes are an important source of SOA that is currently missing from most model representations of organic aerosol formation. This is particularly pertinent given ongoing questions around the applicability of SOA chamber experiments, and hence the derived SOA yields, to real-world conditions.(Kenagy et al., 2024)

Aside from the importance of accretion reactions in remote forested regions highlighted here, recent work has highlighted the potential importance of low-NO RO<sub>2</sub> oxidation pathways (such as accretion reactions) in extremely polluted environments, with high O<sub>3</sub> concentrations.(Hamilton et al., 2021; Mayhew et al., 2023; Newland et al., 2021) Conversely, as NO<sub>x</sub> pollution levels decrease in many regions due to tighter regulations, and BVOC emissions increase due to warmer temperatures and afforestation efforts, RO<sub>2</sub> accretion reactions will likely become a more important source of SOA and could represent a negative climate feedback.

Analysis of the contributions of global average VOC precursors to aerosol-phase accretion products highlights the dominance of biogenic VOCs to accretion product formation, providing support to previous work that has focused only on BVOC RO<sub>2</sub> species. However, our results demonstrate that aromatic species can also be an important precursor in polluted urban environments and those impacted by biomass burning, despite their relatively low contribution to gas-phase accretion product concentrations. Furthermore, this work has highlighted the role of small RO<sub>2</sub> (particularly the peroxyacetyl radical) to accretion product formation, including contribution to SOA when combining with an RO<sub>2</sub> from a larger precursor VOC class like monoterpenes and sesquiterpenes.

Analysis of the accretion product structures indicates that non-peroxide species, such as ethers and esters, may constitute the majority of gas-phase and aerosol-phase accretion products. Experimental confirmation of this will require analysis with an analytical technique able to distinguish chemical isomers, unlike the commonly employed CIMS instruments, since the ester and ether products may have the same molecular mass as the traditionally proposed peroxide analogues. (Franzon et al., 2024) Chromatographic methods (such as liquid chromatography coupled to mass spectrometry) have the potential to offer this isomer-specific detection. (Camredon et al., 2010) The different structures of RO<sub>2</sub> accretion products will be important for attempts to represent their losses in future models given the large difference atmospheric lifetime of organic peroxides compared to esters and ethers. The differential reactivity of peroxide, ether, and ester products may also offer a pathway to distinguish the products with mass spectrometric methods under controlled conditions, for example by observing photolysis rates in chamber experiments.

Finally, although the approach taken here is useful for an in-depth investigation into the global significance and formation processes of SOA from RO<sub>2</sub> accretion products, the large size of the mechanisms employed prohibits its general use. This work demonstrates that there is a wide variety of RO<sub>2</sub> species and VOC precursors contributing to accretion product SOA globally, which presents a challenge when considering how to represent this process in a computationally efficient manner. Furthermore, the current highly parameterised approach of representing SOA formation in atmospheric chemical models prevents model-based investigations like these into the chemical drivers behind SOA formation (e.g. RO<sub>2</sub> accretion and/or autooxidation), given the double-counting issues encountered here. Further work is needed to produce a more condensed representation that recreates the PM<sub>2.5</sub> produced here, while still providing a useful level of chemical detail.

# 635 Code and Data Availability






The data and code required to reproduce the figures presented in this manuscript have been made freely available at https://doi.org/10.7278/S5d-80qm-kyjj.

# **Author Contributions**

AWM conducted the modelling work, analysed the resulting data, and wrote the majority of the paper. LF provided RO<sub>2</sub> accretion reaction branching ratios using GECKO-AP. KHB provided the additional sesquiterpene chemistry added to GEOS-Chem and provided guidance on the GEOS-Chem modelling approach. TK provided guidance on the interpretation of the GECKO-AP output. FDLH, CM, and JAT collected the I-CIMS measurements during the SOAS campaign and provided guidance on interpretation of the data. ARR provided guidance on the implementation of RO<sub>2</sub> accretion reactions in chemical mechanism files. JDH provided guidance and supervision across the whole project. All authors provided input and feedback on drafts of the paper.

# **Competing Interests**

Some authors are members of the editorial board of Atmospheric Chemistry and Physics.

# Financial support

This work was supported by funding from the Postdoctoral Fellowship Program in the Wilkes Center for Climate Change and Policy at the University of Utah. The support and resources from the Center for High Performance Computing at the University of Utah are gratefully acknowledged. The computational resources used were partially funded by the NIH Shared Instrumentation Grant 1S10OD021644-01A1.

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
