# Peer review of "The Global Importance of Gas-phase Peroxy Radical Accretion Reactions for Secondary Organic Aerosol Formation"

_EGUsphere, 2025_

## Author Response (AR1)

We would like to thank the reviewers for their comments. We believe that their suggestions have improved the quality of the manuscript, particularly by helping us to better convey the uncertainties associated with our predictions. Below are point-by-point responses to each comment.

**Anonymous Referee #1**

The authors present a comprehensive and methodologically robust study assessing the global contribution of  $RO_2 + RO_2$  accretion reactions to secondary organic aerosol (SOA) formation using the GEOS-Chem chemical transport model. By integrating updated theoretical understanding and incorporating new accretion mechanisms via GECKO-AP, the work offers a timely and valuable addition to the literature on aerosol formation, especially in tropical and low- NOx environments. The manuscript is technically rigorous, and the authors demonstrate clear model measurement comparisons using data from SOAS and GOAMAZON. It is suitable for publication in ACP. However, I have several suggestions to improve the clarity, balance, and robustness of the conclusions.

**Major Comments:**

- Without directly flagging modeling assumptions the abstract and conclusions suggest that  $RO_2$  accretion products make up 30–50% of  $PM_{2.5}$  in some regions. While the model results support this, the claim should be qualified with reference to underlying assumptions (e.g., product yield estimates, volatility assumptions, possible double counting). Use phrases like "may contribute" or "model results suggest up to..." to moderate certainty.

Due to space limitations in the abstract, we have only adjusted the relevant phrase in the abstract by the addition of the word "may", as suggested by the reviewer. As such, line 23-24 now reads "...The results of this work suggest that  $RO_2$  accretion products may comprise 30-50% of particulate matter (PM2.5) in tropical forested environments..."

Line 412-413 has been adjusted to read "As such, given the underlying assumptions used in the model representation, we estimate that accretion products contribute between 30-50% of the total annual average  $PM_{2.5}$  mass..."

The conclusions section doesn't directly reference the 30-50% figure, but we have adjusted line 598-599 to communicate the uncertainty in conclusions with respect to the 5% figure for temperate regions: "Our results suggest that the contribution of accretion products to OA is lower in more temperate and polluted regions, but may still comprise an average of around 5% in the summer time."

-While the text does mention uncertainties (e.g., calibration, partitioning), these could be more systematically discussed. I suggest including a short paragraph in the discussion or conclusion explicitly listing key uncertainties, such as lack of isomer-resolved detection and structural diversity limitations in GECKO-AP.

As requested, we have included a paragraph in the conclusions section outlining some of the key uncertainties.

Line 575-586: "Given the lack of measurements, there are a number of key uncertainties that should be addressed to allow for more accurate future investigations. Firstly, the ability to measure these highly oxidised organic species is hampered by the lack of authentic standards and the svariable calibration factors for species measured using I--CIMS.

Furthermore, the lack of isomer-resolved detection possible with mass spectrometric techniques presents issues when analysing complex mixtures of accretion products. The sensitivity tests described in "2.4 Global Modelling" demonstrate that our results regarding the formation of SOA are reasonably insensitive to gas-phase photochemical losses of accretion products, meaning that a good understanding of the gas-particle partitioning of accretion products is critical to improving the representation of this SOA formation pathway in future investigations. Finally, GECKO-AP includes multiple simplifications due to a lack of experimental understanding and to limit computational complexity that could be improved in future work, namely the decision to only consider peroxide formation and alkoxy decomposition channels, and to only consider alkoxy decomposition for one of the alkoxy radicals involved."

-There is some ambiguity about whether accretion product mass is being added to or replacing existing parameterized SOA mass in the model. Summarize the key point earlier: that empirically derived SOA yields may already include some fraction of accretion product mass. Provide a clearer summary of how this potential overlap was handled and consider including a schematic or table (perhaps in the SI) for clarity.

A sentence was added to Section 2.4 in the Methodology to further clarify this point. Line 240: "The addition of DISOA to OA in GEOS-Chem means that this additional OA is included in the model on top of the OA produced by GEOS-Chem's existing parameterisation (Figure S1). This has potential to result in double-counting if the existing parameterisation already accounts for some fraction of  $RO_2$  accretion product SOA (see "3.2 Contribution to  $PM_{2.5}$  Mass")."

This addition references a new schematic that has been added to the supplementary information as Figure S1, which demonstrates that the additional SOA contributed in addition to the OA from empirically-derived SOA yields.

-GECKO-AP only considers peroxide formation and alkoxy decomposition channels. This is a major limitation that undermines structural diversity of products (e.g., imines, carbonates, or hydroperoxy derivatives). You should explicitly discuss what classes of real-world accretion products are likely being omitted. Quantify how sensitive your  $PM_{2.5}$  results might be to that structural simplification. Acknowledge this limitation and briefly discuss how it may impact modeled volatility and SOA mass.

There are two points in this comment that require separate responses: One on the structural diversity of products, and one on the reaction channels considered in GECKO-AP. We will discuss these two separately below.

Regarding structural diversity, we took the list of existing RO2 species in the MCM and GEOS-Chem as a starting point when developing the mechanisms used in this work and generated product distributions for these radical lists using GECKO-AP. This means that radicals missing from these base mechanisms, most notably the large variety of highly oxidized monoterpene-RO2 formed from autooxidation, are omitted. See our answer to the question on autooxidation below, including added discussion at lines 586-592 of the revised manuscript. With regards to the specific functionality examples raised by the referee, we note that the focus of our manuscript is on gas-phase organic chemistry producing low-volatility products and their contribution to SOA. We are not aware of carbonate ions playing a significant role in gas-phase atmospheric chemistry, let alone radical reactions.

Imines do play a role in gas-phase chemistry, but they form from OH oxidation of amines (e.g. Bunkan et al. 2019, <a href="https://pubs.acs.org/doi/10.1021/jp5049088">https://pubs.acs.org/doi/10.1021/jp5049088</a>, Almeida & Kurtén 2022, <a href="https://doi.org/10.1021/acsearthspacechem.2c00170">https://doi.org/10.1021/acsearthspacechem.2c00170</a>). We are not aware of studies of reduced nitrogen-containing peroxy radicals forming imine accretion products through  $RO_2 + RO_2$  reactions or other mechanisms. Hydroperoxides, on the other hand, are present in our models, as shown in the list of  $RO_2$  species in Table S2. We believe that the referee may be referencing particle-phase processes that fall outside of the scope of this investigation.

Regarding the product channels in GECKO-AP, the dissociation and H-shift channels (i.e. the non-accretion channels in Figure 1) are not considered in the GECKO-AP branching ratios. In principle, this would suggest that we are overestimating the accretion product formation and thus its contribution to SOA mass. However, addressing this has already been attempted in the GECKO-AP code itself, through systematic exclusion of small RO2 + RO2 pairs from the generated lists of accretion products (for which the non-accretion channels are expected to dominate). This decision was informed by an empirically observed size-dependence in the accretion product yields. Small RO2 reactant pairs for which quantitative experimental data exist have low accretion branching ratios (Summarised in Chapter 3 in this PhD thesis: https://helda.helsinki.fi/server/api/core/bitstreams/b0825124-b819-4195-b209-99212c0629cf/content), but experiments on RO2 + RO2 from monoterpene oxidation report large formation rates of mass signals corresponding to ROOR (Berndt et al. 2018, https://pubs.acs.org/doi/10.1021/acs.est.8b02210, Ehn et al. 2014, https://www.nature.com/articles/nature13032). Theoretically we have been able to connect this observation to the non-covalent interaction energy of the alkoxy radical complex intermediate (Franzon 2023, https://pubs.acs.org/doi/10.1021/acs.jpca.3c01890), and the exclusion of small RO2 pairs in GECKO-AP is based on a parametrised representation of these energies (Franzon et al. 2024, https://doi.org/10.5194/acp-24-11679-2024). While the scarcity of quantitative branching ratio data makes it difficult to draw definitive conclusions, the combined observations made in all of these sources suggests that the combined branching ratios of the peroxide and alkoxy decomposition channels ought to be close to 100 % for the monoterpene-sized RO2 that dominate the particle-phase accretion products in Figure 11. As such, we don't expect the compromises made in our representation of RO2 + RO2 product channels to substantially change the conclusions regarding SOA mass.

The following clarification has been made to the manuscript at line 99-109 in response to this comment, and a similar point raised by Reviewer 2:

"Previous experimental data has shown that the yield of accretion products increases significantly with the size of the reactant RO2 species. (Berndt et al., 2018; Frandsen et al., 2025) This can be explained by the increasing strength of the non-covalent interactions in the intermediate alkoxy radical complex (Figure 1) suppressing the dissociation to free alkoxy radicals. (Franzon, 2023; Franzon et al., 2024) There is no good parametrisation for the yield of the H-shift channel, but a negative correlation between computed H-shift rates and intermolecular interaction energies suggests it can be neglected for sufficiently complex RO2 reactant pairs. (Hasan et al., 2023) Based on these observations, GECKO-AP was designed to estimate this interaction energy for each reactant pair based on the functional groups present, to systematically exclude all weakly-interacting reactant pairs, and to only consider the peroxide and alkoxy decomposition channels for the remaining, strongly interacting reactant pairs. Notably, all pairs containing the common methyl peroxy radical

(CH3O2) are treated as weakly interacting in this parametrisation, and so are excluded from RO2 accretion reactions."

-The manuscript lacks a sensitivity test where accretion products are assumed to have lower yields (i.e., uncertainty in GECKO-AP branching), evaporate faster (i.e., higher volatility), decompose photochemically. Include at least one sensitivity simulation testing either a lower dimer yield (e.g., 50% reduction), or increased loss rate (photolysis / fragmentation surrogate), and assess how much  $PM_{2.5}$  mass this removes globally. This will add robustness and credibility to the 30–50% claim.

We have tested the sensitivity of our results to changing the photochemical loss rate of accretion products from the 1×10-11 cm³ molecule-1 s-1 value used in the original manuscript. We tested changing the rate of this gas-phase OH loss by one order of magnitude in either direction. This results in an OH loss rate constant of 1×10-12 cm³ molecule-1 s-1, which is consistent with the OH reaction of less reactive VOCs like butane or methanol, and 1×10-10 cm³ molecule-1 s-1, which is consistent with the OH reaction of more reactive VOCs like isoprene. The results from these two sensitivity tests are shown in Figures S2 and S3, showing the concentrations of gas phase and aerosol phase accretion products, respectively. The tests demonstrate that while gas phase concentrations of accretion products are sensitive to such large changes in the OH loss rate, the aerosol phase accretion product concentrations are remarkably insensitive to such changes.

Regarding the sensitivity to volatility predictions, we would like to note that the uptake coefficient parameterisation used in this work means that species with sufficiently low volatility will be assigned an uptake coefficient of 1, even if the prediction of saturation vapour pressure varied by orders of magnitude. Similarly, sufficiently volatile products will not undergo uptake in the model regardless of variations in volatility prediction.

A description of the sensitivity tests have been added to Lines 254-264: "A sensitivity test was performed to assess the impact of the selected OH reaction rate on the formation of accretion products. Additional simulations were performed with the accretion product + OH reaction rate constant set to  $1\times10^{-12}$  cm³ molecule-1 s-1 and  $1\times10^{-10}$  cm³ molecule-1 s-1. Each of the simulations were spun up for one year, as with the previously described simulations, and the predicted gas- and aerosol-phase accretion product concentrations for the month of January 2013 are shown in Figure S2 and Figure S3, respectively. These sensitivity tests illustrate that the gas-phase accretion products are sensitive to the change in the OH rate constant, varying by over 600ppt between the two orders of magnitude spanned in the sensitivity test (Figure S2). Conversely, the aerosol-phase accretion products only show a relatively small change in concentration of up to around 0.2  $\mu$ g m-3 between each simulation (Figure S3). This demonstrates that the choice of OH oxidation rate constant has little impact on the majority of the conclusions regarding SOA formation in this work, since the dominant loss of low-volatility gas-phase accretion products is to aerosol uptake, rather than photochemical loss."

We have also mentioned the sensitivity test in the new addition at Line 579: "The sensitivity tests described in "2.4 Global Modelling" demonstrate that our results regarding the formation of SOA are reasonably insensitive to gas-phase photochemical losses of accretion products, meaning that accurate volatility predictions and a good understanding of the gas-particle partitioning of accretion products is critical to improving the representation of this SOA formation pathway in future investigations.".

-"Mean model/measurement ratio was 4.6..." This is substantial overprediction. The explanation (sensitivity-based calibration and fragmentation losses) is valid but not quantified. Provide a range of plausible "true" concentrations using a spectrum of calibration sensitivities (e.g., ± order of magnitude). Consider reporting normalized root mean square error (NRMSE) or similar metrics.

The value of 4.6 is the mean campaign-average ratio for each of the individual signals identified. That is to say that selecting a random measured signal, we would expect the average modelled concentration over the course of the campaign to be 4.6 times higher than the measurement. As explained in the manuscript, accounting for the abundance of each signal brings this average ratio closer to 2 times. We have added the values of the mean ratio, median ratio and NRMSE between the measured and modelled accretion product concentrations to the manuscript text for the reader's information.

Line 325-327: "The mean ratio over the course of the campaign is much higher than 2, at 14.9, because of a large over-prediction between the 25th and 29th of June. The median ratio is 2.17 and the root-mean-squared error, normalised to the mean measured value (NRMSE), is 1.97.".

Since the constant maximum sensitivity calibration is used, the plausible true concentrations range will directly vary in a linear manner depending on the sensitivity offset assumed. For example, assuming that the used calibration is 5 times more sensitive than the real sensitivity would scale the measured concentrations by 5 times. As such, the plausible true concentrations vary from the measured value as a minimum, up to around ten times above the measured values. This consideration motivated our original statement in the manuscript that "this offset of two times is consistent with the maximum sensitivity calibration and is within the range of sensitivity differences for organic species measured by I--CIMS".

We appreciate that the lack of quantification in this section is a limitation, as noted in the manuscript. We regret that we are unable to be more quantitative with this analysis and hope that the additional statistics provided in response to this comment are deemed satisfactory.

-The OA radiative effect changes are described, but without clear error bars or sensitivity runs to support confidence in the conclusion. Add uncertainty estimates (e.g., based on  $\pm 25\%$  OA mass) to the TOA forcing calculations. Even just a bounding box would help.

In order to help strengthen this discussion, we have refined the discussion of the radiative effect over Amazonia to state the calculated average and standard deviation over a defined region (matching the Amazonia region used in Figure 12). The performed sensitivity tests previously discussed should grant some additional confidence in this conclusion since they result in minimal changes to organic aerosol concentrations, and so also show minimal changes in the radiative effect when compared to the base model.

Line 560-561 has been adjusted to read: "...DRE over regions where the  $PM_{2.5}$  increase is large, reaching an annual average decrease of 1.03 W m-2 over Amazonia (latitude-longitude bounding box of (-11.0, -73.0) to (1.0 -57.0)), with a standard deviation of 0.15 W m-2 (Figure 13)."

- Recent literature has shown that peroxy radicals can react on aqueous or organic surfaces (e.g., aerosol interfaces or freshly nucleated particles). These reactions could either enhance or compete with gas-phase dimer formation. Please add a brief discussion addressing surface-phase RO2

chemistry as a competing or complementary pathway. Additionally, could you comment on how including this mechanism might affect your global estimates?

Such processes are not considered in light of the rapid timescale of the RO2 cross reactions that are the focus of this work, and the low collision probability of bulk air molecules with aerosol surfaces in the unpolluted environments where accretion reactions are most important.

Nevertheless, we have mentioned this in a new addition to Line 592, as an example of potential missing  $RO_2$  losses in the mechanisms informing this work: "Furthermore, there are additional  $RO_2$  loss processes not currently represented in the MCM or GEOS-Chem that may compete with  $RO_2$  accretion reactions and reduce their importance relative to the predictions made in this work. For example, recent work has demonstrated the ability for peroxy radicals to undergo surface redox reactions. (Durif et al., 2024)".

**Minor Comments**

- The title could include "SOA" or "secondary organic aerosol" for better visibility.

The title has been changed to "The Global Importance of Gas-phase RO2 Dimerization Reactions for Secondary Organic Aerosol Formation"

- In the Abstract, consider briefly mentioning the distinction between peroxide and non-peroxide products.

Given the limited word count availability in the abstract (the current abstract is at the ACP limit of 250 words), we believe that we would be unable to fully delineate between these two categories in a way that would be satisfactory for a reader without prior understanding of the terms, without removing other important aspects of the abstract.

- Clarify what is meant by "DILVOC" in the radiative effect section and its role in the simulations. Its role in radiative effect calculations is a bit unclear, reiterate its inclusion/exclusion in relevant figures/tables

As stated in the Methodology section, the oxidation products of accretion products (represented by DILVOC) are omitted from all analyses except where explicitly stated (i.e. Figure S10 and the associated discussion). As such, DILVOC is not included in the assessment of radiative effects, as stated in Line 249-251. We have added a sentence to the "Impact on Direct Radiative Effect" section to reiterate this statement.

Line 563: "As discussed in previous sections, there is potential for the oxidation products of gas-phase accretion products to contribute to SOA formation, however these products are omitted from this analysis as DILVOCOA is not passed to the OA mass (see "2.4 Global Modelling")"

- Some figures (e.g., model-measurement comparison plots) would benefit from more detailed legends or axis labels to improve standalone readability without flipping back to text.

In light of this comment and comments from Reviewer 2, we have expanded on the figure captions in Figure 8, 9, S11, and S12.

-Consistently use "RO2 accretion products" or "RO2 dimers" throughout for clarity. Sometimes the manuscript says "accretion products," sometimes "RO2 dimers," sometimes "non-peroxide dimers."

Clarify early that you're referring to peroxide, ester, and ether dimers as the dominant species, and consistently use a single term throughout (e.g., "RO2 accretion products").

In light of this comment and comments from Reviewer 2, additional explanation has been added to the introduction section defining the term "accretion products". Line 40-41: "The dimer products of these RO2 cross reactions are often termed 'accretion products' due to the possibility to form oligomeric products from sequential oxidation.".

We have also added a sentence at Line 49: "The use of the term ' $RO_2$  accretion products' in this work refers to dimer species with peroxide, ester, and ether linkages formed through  $RO_2$  cross reactions.".

We have also gone through the manuscript and ensured that the term 'accretion product' (or similar) is used in place of 'dimer', except where dimer is the more accurate term. This effects lines 126, 249, 314, 318, 500, and 531.

- Must quantify or bound NO biases to ensure accurate RO2 fate modeling.

We agree that NO concentrations are integral to understanding  $RO_2$  fate, and hence the predicted accretion product concentrations. This motivated our inclusion of a discussion of NO concentrations in the SOAS models, as well as extensive illustration of the global  $RO_2$  fate, including proportional loss to NO. If the reviewer has more specific feedback on our discussion of the role of NO on our conclusions, then we are happy to address those comments.

- Need to address potential double-counting of OA when adding new chemistry to existing parameterizations.

We have addressed the impact of double-counting on our conclusions throughout the manuscript, including extensive discussion in Section 3.2, Figure 7, and in the SI. If the reviewer feels that there is a specific instance in which we could better account for the impact of double-counting then we would be happy to address specific comments.

- More clarification is needed on how RO₂ categories like "Small VOCs" and "Mixed VOCs" are chemically defined.

We have added the following description to the main text to compliment the explanation that was already included in the Figure caption for Figure 11.

Line 505: "The explicitly listed precursor groups in this analysis are "Isoprene", "Monoterpenes", and "Aromatics". Any  $RO_2$  from a single VOC that do not fit these categories are classed as "Other VOC". "Mixed VOC" refers to  $RO_2$  that can be formed in the mechanism from more than one of these groups. Finally, due to their diverse sources, any  $RO_2$  with fewer than 4 carbon atoms, regardless of possible VOC source, is placed in a separate category termed "Small VOC"."

- The assumption that nearly 100% of OA is from accretion in some regions is likely an overestimate, needs better constraint or alternative explanations.

The statement that close to 100% of the OA observed during the GOAMAZON campaign is comprised of RO2 accretion products assumes that both the base model's empirically-derived parameterisation and the predicted accretion product concentrations are accurate. Given these assumptions, a doubling in OA concentration on adding the accretion products

implies that the accretion products are 100% double counted and that 100% of the OA is comprised of accretion products.

While the GEOS-Chem OA concentrations are broadly consistent with measured OA during the GOAMAZON campaign, there still appears to be a small over-prediction at ground level, and an underprediction at higher altitudes (Figure S7). Similarly, the uncertainties in the representation of  $RO_2$  accretion products discussed in the manuscript (and clarified in response to this reviewer's comments) allow for this 100% figure to be an over-estimation. The intent of the claims in the original manuscript was not to suggest that 100% of OA in Amazonia is comprised of accretion products, but that our model results suggest that a very large fraction of OA in such environments can be expected to be comprised of accretion products.

We have added some caveats to the manuscript to highlight the specificity of this prediction to the GOAMAZON measurement campaign, the reliance of this conclusion on accurate predictions, and contextualise the 100% figure.

Line 300: "It is likely that the 100% figure is an overestimate given the remaining discrepancies between the base GEOS-Chem OA predictions and the measured concentrations, the uncertainties associated with the predictions of accretion product concentrations, and previous literature exploring other SOA formation pathways in BVOC-rich environments. Nevertheless, this analysis suggests that accretion products comprise a very large fraction of OA measured during the GOAMAZON campaign."

The reference to the 100% figure has been removed from Line 417: "The previous analysis of GOAMAZON data suggests that a large proportion of observed OA could come from accretion products...".

-You should acknowledge how autooxidation might shift results.

We had included a discussion of some potential impacts of autooxidation in the original manuscript, which is at line 362 of the revised manuscript.

To more comprehensively address this, we have added further discussion following the added discussion of model uncertainties at line 586: "The complexity of our representation of accretion reactions precluded the inclusion of autooxidation processes, which could be expected to either increase or decrease the anticipated importance of RO2 accretion reactions. While autooxidation processes will compete with bimolecular RO2 accretion reactions, they also have the potential to form high molecular weight peroxy radicals that could be more likely to undergo subsequent accretion reactions and form low volatility compounds that contribute to SOA. On the other hand, GECKO-AP also includes the potential for hydroperoxide functional groups, which are readily formed by autooxidation, to inhibit accretion reactions through fragmentation to form OH and a closed-shell organic carbonyl species. (Franzon et al., 2024)".

**Anonymous Referee #2**

This article reports the development of chemical models and their implementation into the global EOS-chem model to estimate the importance of RO2 recombination reactions in the formation of low-vapor pressure compounds in the gas and their contribution to SOA globally. It is an interesting study but, in my opinion probably a bit premature because of the scarcity of the experimental data characterizing the reaction pathways of interest here. But I do understand the interest of putting

some figures on the global implications of these reactions, to decide (or justify) to invest more time and resources in this topic.

The results predicting such a global importance for RO2 recombination reactions, even with non-negligible NO concentrations (for instance in the validation in Fig.2) is somewhat puzzling. It would be very interesting to know the rate coefficients used for these reactions in the models. Unfortunately, I could not find this information in the manuscript. This and a few other points regarding the chemical models have prevented me from fully understanding the model results and their implications. These points are listed below, I hope that the authors can clarify them in order to go forward with this paper.

We are pleased that the reviewer recognises the value of this work to motivate further research into this topic, particularly increased measurements under laboratory and field conditions.

Regarding the availability of the mechanisms, they are provided within the code database for the project listed in the "Code and Data Availability" section (<a href="https://doi.org/10.7278/S5d-80qm-kyjj">https://doi.org/10.7278/S5d-80qm-kyjj</a>). The following changes have been made to clarify this:

The mechanism files have been included outside the main directory in the repository, to allow access to them without having to download the large amount of data used to make the figures.

Line 135-136 has been modified to read: "The full modified mechanism file has been made available as supplementary information (see Code and Data Availability)."

Line 141-142 has been modified to read: "All of the mechanisms used in this work have been made available as supplementary material (see Code and Data Availability) in FACSIMILE format for..."

**Detailed comments**

Perhaps the first confusing point is the use of the term "accretion" referring to a range of
compounds without a clear definition. According to the IUPAC gold book this term refers to
all the compounds contributing to aerosol growth. If this is so, and therefore this term
includes all the products discussed in this paper, this should be stated in the introduction.
Otherwise, since the discussion in this paper refers to specific mechanisms, it helps to be as
specific as possible when referring to the products (covalent dimers, ester, ethers...) and/or
mechanism.

The use of the terms ' $RO_2$  accretion' and ' $RO_2$  accretion products' in this manuscript is distinct from the use of accretion to refer to the agglomeration of particles, following their growth in size. We use accretion to refer to a chemical process occurring on the molecular level, as opposed to the accretion of multi-component particles. This is common terminology in the cited literature and is often used as an alternative to "dimerization" since the phrase can be used to refer to oligomeric species in addition to dimers, and the process can also result in the fragmentation of one 'monomer' meaning that the resulting accretion product may contain fewer than the sum of the carbon atoms in the two (or more) contributing  $RO_2$  species.

In light of this comment and comments from Reviewer 1, additional explanation has been added to the introduction section defining the term "accretion products". Line 40-43: "The dimer products of these  $RO_2$  cross reactions are often termed 'accretion products' due to the possibility to form oligomeric products from sequential oxidation. (Hallquist et al., 2009)".

We have also added a sentence at Line 49-51: "The use of the term ' $RO_2$  accretion products' in this work refers to dimer species with peroxide, ester, and ether linkages formed through  $RO_2$  cross reactions.".

2. The text mentions that the full modified mechanisms are available in the Supplementary Material, but I can not find this anywhere. The SI only includes a list of non-accretion reactions. Did I miss something?

As previously noted, all of the mechanisms are included as part of the code database, and changes have been made to clarify this fact.

3. I am not sure to understand the statement li. 94-97 p.4 "However, its prediction of the product distribution is highly uncertain ... For this reason GECKO-AP only considers peroxide formation and alkoxy decomposition channels"... Do the authors mean that the channels in Fig. 1 are the ONLY ones of RO2+RO2 (or RO2 + R'O2) taken into account in GECKO-AP? Or that these are the only added channels because the others are already in GECKO-A? Since I could not find the full mechanisms, I could not check which one it is.

We apologize for the lack of clarity. What is meant by this sentence is that the peroxide and alkoxy decomposition channels are the only product channels of  $RO_2$  +  $RO_2$  considered by GECKO-AP. The alkoxy pair-forming dissociation and alcohol- and carbonyl-forming intermolecular H-shift channels are neglected, though they are included in the final mechanisms owing to their inclusion in the base MCM and GEOS-Chem mechanism. See above for our response to Reviewer #1 on what informed us to make this decision when developing GECKO-AP, and why we do not expect it to have a significant impact on the modelled SOA mass.

We have rephrased the wording at line 99-109 in the revised manuscript to clarify the assumptions made in GECKO-AP:

"Previous experimental data has shown that the yield of accretion products increases significantly with the size of the reactant  $RO_2$  species. (Berndt et al., 2018; Frandsen et al., 2025) This can be explained by the increasing strength of the noncovalent interactions in the intermediate alkoxy radical complex (Figure 1) suppressing the dissociation to free alkoxy radicals. (Franzon, 2023; Franzon et al., 2024) There is no good parametrisation for the yield of the H-shift channel, but a negative correlation between computed H-shift rates and intermolecular interaction energies suggests it can be neglected for sufficiently complex RO2 reactant pairs. (Hasan et al., 2023) Based on these observations, GECKO-AP was designed to estimate this interaction energy for each reactant pair based on the functional groups present, to systematically exclude all weakly-interacting reactant pairs, and to only consider the peroxide and alkoxy decomposition channels for the remaining, strongly interacting reactant pairs. Notably, all pairs containing the common methyl peroxy radical ( $CH_3O_2$ ) are treated as weakly interacting in this parametrisation, and so are excluded from RO2 accretion reactions."

The first case would obviously lead to large overestimations of the recombination products. This would also explain the "excessive and unrealistic build-up of accretion products" by the model mentioned several times in the paper. If a model tends to grossly overestimate some products, it probably means that it is not realistic. Could the authors comment on why they expect the model to overestimate these products unrealistically?

In any case, if a choice has been made to determine upper limits for these recombination products, this should be clearly stated and remined when discussing the model results.

We would like to note that the discussion of "excessive and unrealistic build-up of accretion products" is only in the context of an additional photochemical loss added to the GEOS-Chem simulation, and as such is only mentioned once in the paper. Of course, we do not believe that the model "grossly overestimates" the product concentrations, as supported in the manuscript by reference to the SOAS box model results and comparisons to previous literature. The additional photochemical loss pathway added in GEOS-Chem is a reflection of chemical losses that would be present in the real-world atmosphere. Such additional losses are not required in the box models due to the assumption that the loss of accretion products is dominated by physical processes, including ventilation (i.e. removal of the species out of the 'box'). The sensitivity of our results to this additional photochemical loss in GEOS-Chem has been addressed in response to comments from Reviewer #1.

We also note that even in the case where the models do overestimate the accretion product yield from  $RO_2 + RO_2$  reactions, their formation rates in the simulations are physically constrained by the competition between  $RO_2 + RO_2$  and other reactions of peroxy radicals. As noted in Section 2.1 of the manuscript, we consider the representation of  $RO_2 + RO_2$  rate coefficients in the models less uncertain than the representation of the branching ratios, as the structure-activity relationship used (Jenkin *et al.* 2019 (https://acp.copernicus.org/articles/19/7691/2019/)) performs well against experimental data for a large variety of  $RO_2$  species.

1. Figure 1 is a bit confusing for readers who are not expert in RO2 chemistry. Perhaps the "classical" channels of RO2+RO2 could be reminded (even very briefly) to clarify that only the "third" or other channels are detailed here. Also, numbering some of the pathways (for instance "1" for dimerization, "2" for ether/ester...) for would simplify (and clarify) some discussions in the text.

Figure 1 has been updated to more explicitly highlight the accretion and non-accretion pathways.

2. In the second part of the work, the study of the distribution between peroxides and ester/ether products, a list of the VOC or RO2 involved is missing. Was it the VOC mix for the SOAS campaign? According to the Peräkylä et al., 2023 paper the formation of esters/ethers is specific to RO2 with a carbonyl group in  $\beta$  of the RO2 group. Beside some terpenes, this is not a general feature in RO2, thus the results in Fig. 10 would be highly dependent on the VOC mixture chosen.

As noted in the manuscript and the caption to Figure 10, the presented product distribution is taken from the MCM-Accr box model results for the SOAS campaign, and so uses the measured VOC concentrations for the VOCs listed in Table S1. While

the reviewer is correct that this distribution will vary depending on the VOCs present, we would like to highlight the following three points:

- 1. The VOC mixture in SOAS will be heavily dominated by BVOCs (particularly isoprene), and so is not an unreasonable analogue for the remote forested environments we find to be important globally.
- We can only perform this analysis on the more chemically complex box model output because of the simplifications made to the mechanism when representing the chemistry in GEOS-Chem, hence our analysis focusing on the SOAS box model results.
- 3. While the reviewer is correct regarding the requirement for a  $\beta$ -carbonyl to form ester linkages, these are not the only functionalities facilitating the rapid decomposition of alkoxy radicals and forming ether and ester products. Indeed, a  $\beta$ -carbonyl must not be present for the formation of ether linkages. These structural requirements are accounted for in the predictions made by GECKO-AP and we refer to the discussion in Section 3.1 of Franzon et al. 2024 on when GECKO-AP predicts high yields of ethers and esters.

We have added a statement at Line 486 to emphasise that the accretion product composition will be dependent on the VOC mixture present in any given environment, including reference to a new supplementary figure showing the average diurnal mixing ratio of constrained VOCs in the MCM box models: "The structure of accretion products will depend on the structure of the contributing peroxy radicals, which in turn depends on the VOC mixture present in a given environment. The average diurnal mixing ratios of constrained VOCs during in the MCM box model simulations are shown in Figure S16."

3. Although the above questions prevented me from fully appreciating the modelling results, I have a few questions on these results:

- In the validation presented in Fig. 2, the "dip" in the products between 5 and 10 am (especially in the gas-phase products) is clearly correlated with the NO peak presented in Fig. S1 and underestimated by GEOS-Chem. But, at all the other times, measured and predicted NO agree well. Yet, the GEOS-Chem simulation overestimates the products over all these times (for instance around 15:00 for the gas-phase ones). Thus, shouldn't there be other explanations for this overestimation of the products by GEOS-chem than the discrepancies between the measured and predicted NO?

While there is a period of agreement between the measurements and GEOS-Chem in the afternoon period, the NO underprediction extends throughout the night-time period and into the morning. The magnitude of this NO underprediction is not necessarily directly linked to the magnitude of the accretion product discrepancy, as the RO2 fate and cumulative formation of accretion products will impact this discrepancy.

The night-time underprediction of NO is central to the second half of our explanation at Line 287-289, where high night-time  $NO_3$  concentrations facilitated by lower NO results in higher night-time  $RO_2$  concentrations (Figure S2). The lifetime of the accretion products and the combination of these two effects can account for the higher background concentration of accretion products, with the diurnal trends in NO over the course of the day modifying the background concentration and producing a morning dip in concentrations.

As mentioned above, the substantial amount of recombination products predicted by the model, even over 10 - 17 h when the NO concentration is  $\sim 0.05$  ppb is puzzling. It seems that RO2 + NO should be much faster than RO2+RO2 under these conditions. But, of course, it depends on the rate coefficients assumed for RO2+RO2, which brings back to question 2 above.

While these NO concentrations may result in competitive NO +  $RO_2$  reactions, especially for specific  $RO_2$  species, we would suggest that 50 ppt of NO should generally be considered low concentrations of NO. As such, it is not surprising to us that we see evidence of  $RO_2$  accretion reactions in the SOAS region. This is particularly true in light of the fact that the SOAS region shows relatively low amounts of accretion product formation compared to the much more remote forested regions highlighted by the global simulations, as noted in the manuscript.

At 1 atm of pressure and a temperature of 300 K, 50 ppt of NO corresponds to  $\approx 1.0 \times 10^9$  molecules cm-3, which is comparable to the  $\approx 0.8 \times 10^9$  molecules cm-3 of total RO2 at the same time in our box models (Figure S5). According to Jenkin *et al.* 2019 (https://acp.copernicus.org/articles/19/7691/2019/), the rate coefficients for RO2 + NO are roughly comparable to the most rapid RO2 + RO2 reactions, implying that some but not all RO2 + RO2 reactions will be competitive under these conditions.

Furthermore, according to Newland *et al.* 2021 (<a href="https://doi.org/10.5194/acp-21-1613-2021">https://doi.org/10.5194/acp-21-1613-2021</a>), under conditions experienced during the SOAS field campaign and at NO concentrations of 50 ppt, we should expect a maximum of around 50% of a reference isoprene  $RO_2$  to be lost to reaction with NO, and potentially less than 25% depending on the  $RO_2$  and  $HO_2$  concentrations present. This therefore leaves plenty of additional reactivity to be accounted for by RO2+HO2 and RO2+RO2 reactions. Indeed, even at the peak average morning NO concentrations of 0.3 ppb, we can expect 50-90% of  $RO_2$  to be lost to reaction with NO, leaving room for accretion product formation.

- The results in Fig.8 and 9 showing major losses of RO2 at very high (fig. 8d) and very low (Fig. 8e, 9b, 9d) latitudes are rather surprising and need to be discussed. How can there be major RO2 losses at these high latitudes, where the RO2 concentrations are expected to be small anyway (few VOC emitted + lack of light for 6 month of the year)?

Figure 8b-f and Figure 9 show the proportional loss of a representative  $RO_2$  to each pathway, meaning the presented value is independent of the  $RO_2$  concentrations. The reviewer is correct that these environments likely contain very low  $RO_2$  concentrations, and this is reflected in Figure 8a which shows the total IHOO1 loss rate (which *does* depend on the IHOO1 concentrations) in these regions to be very small.

We have made the following changes in order to clarify this for the reader.

Line 439-442: "Given that the data in panels (b-f) are expressed as a proportional loss, the values are independent of  $RO_2$  concentration. Figure 8a shows the total loss rate taken from the model, which is dependent on the concentration of the  $RO_2$  and any reaction partner (e.g. NO,  $HO_2$ , and  $RO_2$ ), illustrating regions in which the  $RO_2$  chemistry is most globally important."

Figure 8 caption: "(a) Average total loss rate of the primary isoprene  $RO_2$ , IHOO1, during June of the GEOS-Chem simulation. (b-f) Average fractional loss of IHOO1 to each loss pathway, independent of  $RO_2$  concentration."

Figure S11 caption: "(a) Average total loss rate of the primary isoprene  $RO_2$ , IHOO1, during December of the GEOS-Chem simulation. (b-f) Average fractional loss of IHOO1 to each loss pathway, independent of  $RO_2$  concentration."

Figure 9 caption: "Proportional loss of  $RO_2$  to accretion reactions, independent of  $RO_2$  concentration, during June of the GEOS-Chem simulation. (a) average for all  $RO_2$ , (b) primary  $RO_2$  from styrene oxidation (C2BZRO2), (c) primary  $\alpha$ -pinene  $RO_2$  (APINO2), (d) primary sesquiterpene  $RO_2$  (SQTO2)."

Figure S12 caption: "Proportional loss of  $RO_2$  to accretion reactions, independent of  $RO_2$  concentration, during December of the GEOS-Chem simulation. (a) average for all  $RO_2$ , (b) primary  $RO_2$  from styrene oxidation (C2BZRO2), (c) primary  $\alpha$ -pinene  $RO_2$  (APINO2), (d) primary sesquiterpene  $RO_2$  (SQTO2)."

---

## Author Response (AR2)

We thank the reviewer for their comments, below are point-by-point responses referencing the adjusted manuscript. The line number reference line numbers in the "track changes" document.

**Major comments:**

1. I suggest the title to be changed to "The global Importance of Gas-phase Peroxy Radical Accretion Reactions for Secondary Organic Aerosol Loading" or "......Secondary Organic Aerosol Budget". The term "formation" suggests an emphasis on the process, and therefore I thought I would read a lot of content in nucleation, aerosol growth etc...but what the paper primary studies is the impact on aerosol mass and radiation.

As requested, the title has been changed to refer to 'Aerosol Loading' as opposed to 'Formation'.

2. One important piece of information I did not find is how accretion product signals in CIMS were selected. Are they ions with more than 10 carbon? Besides, what signals contribute to the spike at 5 am in Figure 3, and can those species provide insights in the potential source of model-measurement disagreement? Is it possible to do some sort of species-to-species or groups of species comparison with MCM-Accr, even just on molecular formula? There may be some useful information that haven't been sufficiently utilized in box modeling to address some of the uncertainties and characterize the importance of reaction pathways.

The means by which accretion product signals were selected is outlined in Section 2.2. Specifically, the reported ion masses were compared against species from the MCM-Accr mechanism, and any model masses for which accretion products made up an overwhelming fraction of the mass were selected for comparison. This certainly omits some measured accretion products, but the aim of the filtering was to avoid comparison with signals that may include large amounts of non-accretion products which are not accounted for in the box model's aerosol partitioning calculations.

Regarding the signals contributing to the measured I-CIMS signal, we have produced the figure below for the interest of this reviewer. As can be seen, the average contribution of the top 9 measured signals (from the subset selected for the accretion product comparison) shows little variation over the course of the day, including during the 5 am spike.

While it would be interesting to investigate trends in individual signals further, it is difficult to make specific comparisons given the uncertainty of sensitivities for individual species/signals. This is why we limited our discussion in the manuscript to demonstrate

that the bulk accretion product concentrations of the model are in-line with the measurements. As noted in previous review responses, we regret that we are not able to be more quantitative in this section but believe that this paper should act as motivation for increased and updated measurements to inform future model representations.

3. Did the AMS use a PM2.5 inlet in the field campaigns? The authors may want to clarify, because some AMS measure PM1, some measure PM2.5, and this may affect the analysis results here.

Both of the AMS datasets used (Amazonia and South-eastern USA) measured sub-micron aerosol. This has been clarified in the manuscript by the addition of the following text:

Line 165 has been changed to read: "Submicron OA measurements from this campaign, measured by aerosol mass..."

Line 168 has been changed to read: "...the AMS measured submicron OA during the GOAMAZON campaign."

4. In Figure 2 and 3, the diurnal variation in SOAS seems stronger in modeled results than in measurements. In the modeled results, the peak concentration at night can be more than 3 times higher than day-time concentration, but such contrast is less in measurements. What could be the possible reason?

It should be noted that Figure 2 compares the box model results to the GEOS-Chem output, not measured data. As such, the reviewer is instead noting that the box models produce a stronger diurnal than both the GEOS-Chem output and the I-CIMS measurements.

The diurnal profile will be sensitive to the representation of physical processes in the box model, such as varying boundary layer mixing, which can strongly influence species concentrations at the surface. Such physical processes are difficult to represent in box models, particularly for long-lived stable species. We have attempted to represent such processes in this work by accounting for both deposition (scaled by boundary-layer height) and ventilation (scaled based on a reference compound of MVK), which is more complex than the approach taken in many ambient box modelling investigations and has been validated in previous work (Mayhew et al. 2022). Nevertheless, it is not unexpected that the box models are unable to capture the diurnal profile perfectly because of the complexity of representing physical processes.

This has been noted in the paper by adding the following text. This text focuses on the comparison to measurements, since boundary layer dynamics have been shown to be a source of error in GEOS-Chem and so it is best to not rely on the GEOS-Chem output as "truth" when focusing on relatively small discrepancies in the diurnal profile.

Line 332: "The diurnal average profile of the measured and MCM-Accr data show similar trends, with higher night-time concentrations. However, the modelled diurnal is stronger than that observed with I-CIMS. This may suggest that the representation of physical processes in the box model are either removing species to quickly during the day, or not quickly enough at night, particularly for aerosol-phase accretion products."

5. In Figure 2 and S4, the authors attribute the higher accretion products in GEOS-Chem primarily to the overprediction of NO. In Figure S4, GEOS-Chem did a nice job after 10 am. However in Figure 2b, GEOS-Chem's prediction is still higher by several factors (from 10am-5pm for example). This suggests that besides NO, there are other important drivers for the discrepancy that need to be discussed.

The reviewer is correct that the GEOS-Chem simulation offers good predictions of NO in the afternoon period, which does not correspond to accretion product concentrations in line with the box models in Figure 2. However, given the lifetime of the produced accretion products, the increased production from lower NO concentrations (resulting in more  $RO_2+RO_2$  reactions and higher night-time  $RO_2$  concentrations) will produce accretion products that are able to persist over the course of the day. Indeed, the OH loss rate constant of  $1\times10^{-11}$  cm3 molecules-1 s-1 used in the GEOS-Chem simulations results in a chemical lifetime of over 1 day (assuming [OH] =  $1\times10^6$  molecules cm-3). As such, we maintain that higher production at a given time of day can explain the higher concentrations observed over the course of the whole day.

We do also already mention the potential influence of differences in the representation of physical processes between the box models and GEOS-Chem as an additional source of the difference.

6. If the underprediction of NO by GEOS-Chem is (one of) the main reason for overpredicting the accretion products in SOAS, how is this issue in NO underprediction (or NO simulation uncertainty more generally) addressed later when assessing the global-scale impacts of adding accretion products?

The reviewer is right that there may be errors in GEOS-Chem's representation of NO (both under and over-predictions) that could affect the observed accretion product formation. However, GEOS-Chem's ability to simulate NO concentrations is not directly addressed as part of this investigation as there may be numerous causes of poor NO predictions in different global regions, including both emissions and secondary chemical processes shifting  $NO/NO_2$  partitioning.

In recognition of this limitation, we have added the following statement to the conclusion.

Line 614: "The representation of NO concentrations in global models like GEOS-Chem is important for accurately representing  $RO_2$  loss pathways, as illustrated by the SOAS model results presented here. Uncertainties in global NO predictions could act to either increase or decrease the presented global accretion product concentrations. Future efforts to produce more mechanistic representations of SOA formation will require confidence in the predictions of gas-phase atmospheric species such as NO and  $HO_2$  through their continued validation against measurements collected over long periods in a range of environments."

7. I agree with one of the previous reviewers that the accretion reactions used in the mechanism should be given. Although the authors provided full reactions in "Code and Data Availability", it would be very nice if the authors can provide a subset of the reactions (for example, even just for one key species) in the SI. This will give the readers ideas on how the chemistry is treated. If the readers are interested, they can then dive deep into the full mechanism.

Given the sizes of the mechanisms produced, it is difficult to summarize them in a way that is both succinct and comprehensive. However, to give the reader an idea of the implementation approach in response to this comment, we have included a handful of examples in a new SI table (Table S4). This table is referenced at Line 144:

"A small selection of reactions has also been listed in Table S4 to demonstrate the implementation."

**Minor comment:**

8. Please explain "LVOC" when it first appears in Line 147 (instead of in line 257).

The explanatory sentence has been moved, as requested.